# Evolutionary regulation of human Fas ligand (CD95L) by plasmin in solid cancer immunotherapy

Brice E. N. Wamba[1,2], Tanmoy Mondal[1,2], Francis Freenor V[1,2], Mehr Shaheed[1,2,3], Oliver Pang[1,2,3], Daniel Bedinger[4], Patrick Legembre[5], Laurent Devel[6], Sanchita Bhatnagar[2], Gary Scott Leiserowitz[7] & Jogender Tushir-Singh [1,2,7,8] ✉

Despite sharing >98% genomic similarity, humans are more likely to develop cancers than our closest living ancestors, the nonhuman primates. Here, we unexpectedly discover that, unlike chimpanzee and other primates, a critical embryonic development, immune homeostasis, and general cell-death regulator protein called Fas Ligand (FasL) contains a Pro153-Ser153 evolutionary substitution in humans. The latter renders human FasL preferentially susceptible to cleavage by plasmin, an overly elevated protease in solid tumors. Since FasL-mediated killing of tumor cells by activated T-lymphocytes and chimeric antigen receptor T-cells (CAR-T) is critical for therapeutic efficacy, we find that elevated plasmin levels in certain ovarian tumors interfere with the T-lymphocyte-expressed FasL death signaling. Either targeted inhibition or blocking plasmin accessibility to membrane FasL rescues the FasL cell-death function of activated T-lymphocytes in response to immune-checkpoint receptor targeting antibodies. These findings of evolutionary significance highlight that elevated plasmin in metastatic tumors potentially contributes to differential outcomes of T-cell-based immunotherapies in solid tumors.

Clinical data indicate that antibodies that help eliminate tumor cells by activating and harnessing the power of a patient's immune system are highly effective against hematological cancers[1]. The core of immunotherapy primarily relies on strategies that potentiate, prolong, and maintain T-cell function, either using T-cell activating monospecific and bispecific antibodies against checkpoint blockade receptors or via chimeric antigen receptor (CAR)-T cells[2]. Regarding clinical response rates, T-cell-based immunotherapies, unfortunately, have a significant disparity in hematological liquid vs solid tumors[3]. On the one hand, various regulatory mechanisms intrinsic to T-cell priming and activation are influenced by solid tumor microenvironment (TME)[4]. At the same time, the tumor's inherent factors contributing to limited

efficacy include the loss and downregulation of target antigen (antigen escape), resulting in high tumor heterogeneity, further influencing the outcome[5]. Moreover, before everything, limited T-cell and general immune penetration remains a significant bottleneck in dense solid tumor stroma. Hence, alternate strategies that debulk solid tumors independent of immune effector cells are highly critical.

For the past few decades, most alternatives have focused on targeting death receptor-activating agonist antibodies, as the latter has the potential to shrink tumors via extrinsic apoptotic cytotoxicity independently of limitedly solid tumor stroma penetrated immune effector cells[6]. TRAIL-R2/DR5 remains the essential targeted TNFα-superfamily death receptor in various solid tumor trials; however,

[1]Laboratory of Novel Biologics, University of California Davis, Davis, CA, USA. [2]Deartment of Medical Microbiology and Immunology, University of California Davis, Davis, CA, USA. [3]Undergraduate Research Volunteers Program, University of California Davis, Davis, CA, USA. [4]Carterra, Salt Lake City, UT, USA. [5]UMR CNRS 7276, INSERM U1262, University of Limoges, Limoges, France. [6]CEA, INRAE, Médicaments et Technologies pour la Santé (MTS), SIMoS, Université Paris-Saclay, Gif-sur-Yvette, France. [7]Department of Obstetrics and Gynecology, UC Davis School of Medicine, University of California Davis, Davis, CA, USA. [8]UC Davis Comprehensive Cancer Center, UC Davis School of Medicine, University of California Davis, Davis, CA, USA. ✉e-mail: jtsingh@ucdavis.edu

tested DR5 agonists have shown limited clinical success[7]. More recently, TNFα-superfamily death receptor named Fas (CD95) has become the focus of attention due to the critical role of Fas ligand (FasL) on the surface of CAR-T cells in instigating bystander killing of antigen⁻/FasR⁺ cancer cells in leukemia clinical trials[8–10]. The bystander killing is critical for CAR-T and other T-cell-based immunotherapies against antigen-escaped tumors[11]. Hence, circumstantially, a higher FasL signaling against the tumors in TME, independent of its activity by tumor-infiltered T-cells (in response to immunotherapies), natural killer cells, stromal cells, or engineered CAR-T cells should indicate a better antitumor outcome[8–10]. Contrarily, despite the higher FasL/Fas mRNA and protein ratio, the prognostic effect of Fas signaling remains inconclusive in tested solid tumor studies[12–14]. Besides, the differential and temporal regulation of FasL-mediated extrinsic apoptosis is also a hallmark of the T-lymphocyte self-suicide program[15] in the spleen and lymph nodes. Indeed FasL/Fas signaling is critical to maintaining immune homeostasis and is solely responsible for monitoring lymphoproliferative disorders[16]. The latter indicates a discrepancy in operating optimal FasL signaling in healthy organs such as in spleen and lymph nodes vs aggressive solid tumors. A key difference between T-cell homeostasis and metastatic solid tumors is upregulated proteolysis[17,18], which degrades the extracellular matrix and promotes metastasis[19]. Hence, in this work, we sought to investigate whether distinctive regulatory mechanisms predispose FasL to differential proteolytic degradation in tumors and if targeting the latter could improve T-cell-based immunotherapies.

## Results

### Proline 153 to Serine153 substation in human FasL vs nonhuman primates

The proteolytic degradation of tumor suppressor TP53 by HPV-E6 oncoprotein is a well-studied model for allelic polymorphism and evolutionary variations in humans[20,21]. For example, p53, having Pro72 vs Arg72 is significantly resistant to degradation by HPV-E6[20,21]. Strikingly, all non-human primates contain Pro72 in p53 except the human lineage. Similarly, when compared between humans and nonhuman primates, a critical substitution of Ser153 instead of Pro153 is evident only in human FasL (Fig. 1a). Till today, no other study has analyzed the implication of Pro153Ser mutation on FasL, and a FasL[S153P] single nucleotide polymorphism (SNP:rs2101810509) is a rare variant in human dbSNP database (Supplementary Fig. 1a-d).

At the FasL structure level, Ser153 is present on beta-strand A-A' connecting loop[22] (Fig. 1b). Notably, the closest protease regulatory site in close proximity to Ser153 is the plasmin cleavage loop ($^{144}RK^{145}$) on the other side of the same "A" β-strand (Fig. 1b). The S153P substitution brings proline closer to the aromatic ring of F269 (2.6 Å) in the neighboring H-strand loop to potentially stabilize FasL extracellular domain (ECD) differently than humans via hydrophobic and CH/π interactions or π stacking (F269-P153)[23] (Fig. 1c). Considering that CH/π interactions are essential for protein structure and function[24], we hypothesized that the S153P substitution in FasL (unlike S153) could potentially alter the "A" β-strand conformation to interfere with optimal plasmin accessibility or cleavage at the $^{144}RK^{145}$ residues. To test our hypothesis, we first cloned and purified the N- and C-terminal His-tag entire ECD of human and Rhesus monkey FasL (RhFasL) using CHO cells. Unfortunately, C-terminal His-tag version of huFasL and RhFasL were poorly expressed and were not functional as compared N-terminal tagged proteins (Supplementary Fig. 2a-f, also see discussion). Strikingly, RhFasL ECD showed a higher trimer/monomer ratio than huFasL (Fig. 1d), indicating potentially slightly differential conformations of HuFasL and RhFasL ectodomains.

### Human FasL is selectively targeted by plasmin at a single specific site

If metalloproteases (MMPs) or ADAM10 cleavage of membrane-bound FasL in its stalk region (such $^{110}EL^{111}/^{113}EL^{114}/^{129}KQ^{130}$) could be influenced by P153S substitution at the end of stalk[25], we incubated purified his-tagged huFasL and RhFasL with recombinant adam10 as published[26] followed by immunoblotting with anti-FasL antibody. Both HuFasL and RhFasL were equally sensitive to ADAM10 (Fig. 2a). Next, we tested if evolutionary P153S substitution would influence the previously described[22] cleavage of human vs primate FasL at $^{144}RK^{145}$ position. Strikingly, only huFasL was highly susceptible to plasmin (Fig. 2b-d, Supplementary Fig. 3a-b). It must be noted that the kinetic binding behavior to the immobilized human plasmin (covalently linked to an HC30M liner polycarboxylate chip surface) was almost equivalent for the huFasL and RhFasL (Fig. 2e, Supplementary Fig. 4, 5). The latter suggests that cleavage insusceptibility of RhFasL is not due to S153P mutation interfering with the plasmin engagement loop (PEL) near $^{144}RK^{145}$ residues. Next, we transiently expressed myc-tag full ECD of human and rhesus membrane FasL in CHO-K cells to confirm if exogenously added plasmin (Pln) will cleave membrane expressed FasL ECD (Fig. 2f). Similar results were seen where huFasL was highly sensitive compared to rhFasL (Fig. 2g). Moreover, preincubation of huFasL (but RhFasL) with plasmin followed by addition on tumor cells abolished its cell-killing and caspase-8 activation function (Fig. 2h-i, Supplementary Fig. 3c). Furthermore, adding plasmin directly in culture media after recombinant FasL treatment also resulted in selective huFasL (but RhFasL) cleavage (Supplementary Fig. 3d).

Unfortunately, the immunoblotting signal of his-tagged FasL was completely lost after plasmin cleavage (Fig. 2b, d). The latter might indicate two possibilities: a) the presence of multiple plasmin cleavage sites in FasL, or b) the anti-FasL antibody epitope being compromised by the plasmin cleavage. To overcome these issues, we tested anti-His-antibody for immunoblotting; however, other than having multiple non-specific bands, we also had difficulty detecting the plasmin-processed extremely short-fragment (~40 amino acids) after cleavage at $^{144}RK^{145}$. Hence, we cloned and expressed FasL as an IgG1-Fc conjugated protein. Specifically, the entire FasL ectodomain (Q103-L281 aa) was genetically ligated to the C-terminal of CH3 domain of a random IgG1 Fc via G4S linker (Random IgG1-FasL$^{103-281}$) (Fig. 3a, Supplementary Fig. 6a). The latter format (VH-CH1-CH2-CH3-linker-FasL$^{103-281}$) we have previously shown[10] to be functional and effective. On the contrary, when FasL ectodomain (Q103-L281 aa) was genetically ligated to the N-terminal of IgG1-hinge followed by Fc (FasL$^{103-281}$-hinge-CH2-CH3), it was not functional (see "Discussion"). We also engineered an optimal plasmin cleavage site in the IgG1 hinge region in such a way that IgG1-FasL now contains two plasmin cleavage sites: one in IgG1 hinge and the other in $^{144}RK^{145}$ FasL (Fig. 3a, sequences are shown, and the scissors represent plasmin cleavage sites). Compared to a regular IgG1 heavy chain of ~50 kDa, the IgG1-FasL heavy chain would be 71 kDa (See box "A", in Fig. 3b). If plasmin cleavage happened only at the IgG1 hinge (see schematic Scissor 1), an exactly 47kDa fragment with CH2-CH3-linker-FasL will be released (See box "B" in Fig. 3b). If plasmin cleavage happened only at the FasL site (see schematic Scissor 2), another exactly 47 kDa fragment with VH-CH1-CH2-CH3 and a small linker will be released (See box "C" in Fig. 3b). If plasmin cleavage happened at both places (see a schematic of Scissor 1 and Scissor 2), a ~31 kDa fragment with CH2-CH3-small linker containing fragment will be released (See box "D" in Fig. 3b). The latter (~31 kDa fragment) will decisively indicate the cleavage of FasL at $^{144}RK^{145}$. Importantly, all released fragments, plus intact 71 kDa heavy chain, would contain intact Fc (CH2-CH3) and be easily detected on SDS-immunoblot by anti-human Fc antibody. Notably, only IgG1-FasL that had huFasL (linked to CH3 domain) released a highly prominent ~31 kDa fragment (in a time course dependent manner), further reaffirming selective significant cleavage of huFasL by plasmin (Fig. 3c and Supplementary

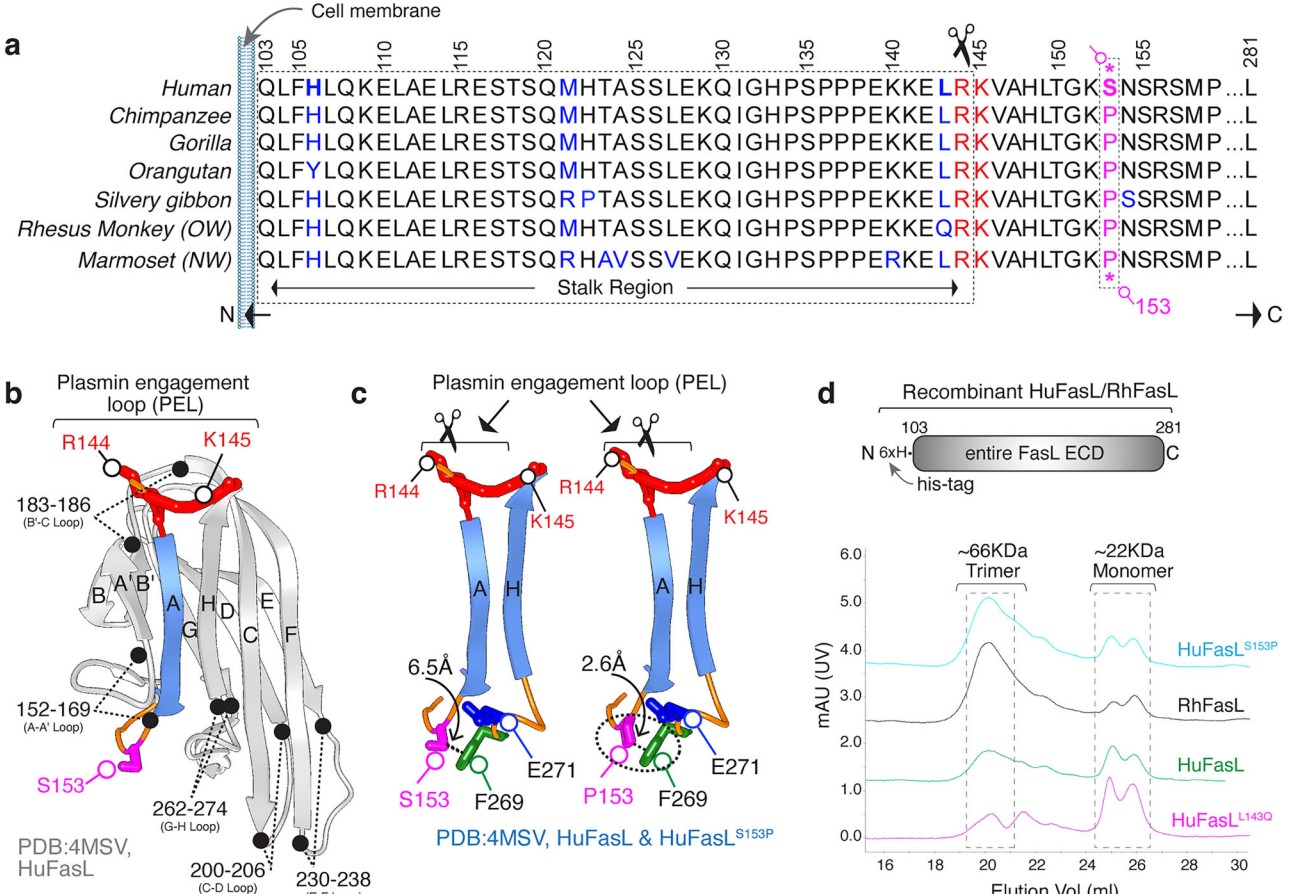

**Fig. 1 | Proline to serine substitution at position 153 in HuFasL. a** Amino acid alignment of the FasL membrane proximal ECD from different primates. **b** Ribbon depiction of FasL monomer. **c** FasL A and H beta-strands are highlighted along with their connecting loops. R144, K145 (red), S153/P153 (left/right red), F269 (green), E271 (blue) are shown as sticks. **d** SEC chromatograms of indicated FasL are shown with different colors.

Fig. 6b with the ladder shown). Next, to conclusively confirm that [144]RK[145] residue is the only plasmin cleavage site in close vicinity of evolutionary S153 mutation, we generated another IgG1-huFasL with [144]AA[145] mutation and tested in the same assay. As evident in Fig. 3d (and Supplementary Fig. 6c with the ladder shown on right blot), [144]AA[145] mutation abolished ~31 kDa fragment release. Furthermore, P153S substitution in IgG1-RhFasL highly sensitized it to plasmin cleavage (Supplementary Fig. 6d). Similar substitution-driven sensitivity and resistance results were seen with the His-tagged huFasL and RhFasL (Supplementary Fig. 6e, f).

### Mouse FasL is targeted by plasmin at sites close to the membrane and is different than human FasL

Unlike RhFasL, the murine FasL (muFasL) sequence is not highly conserved with human FasL in the stalk region as well as in the entire ectodomain (Fig. 4a, Supplementary Fig. 7a). The corresponding plasmin cleavage region [143]LRKV[146] is [143]PRSV[146] and [152]KSN[154] is [152]NPH[154] in muFasL (Fig. 4a). Next, we generated IgG1-muFasL and tested in plasmin cleavage assay. Surprisingly, the plasmin cleaved fragment was 27 kDa rather than 31 kDa (Fig. 4b, Supplementary Fig. 7b: actual ladder shown), suggesting cleavage site not being at RS of [143]PRSV[146] [14]. A shorter 27 kDa fragment release indicated the cleavage site being different and being close to transmembrane domain rather than at [143]PRSV[146] (Fig. 4a). We repeated the experiment by replacing the random IgG1 backbone with anti-FOLR1 IgG1 framework (Supplementary Fig. 7c). Similarly, no 31kDa was released confirming the [143]PRSV[146] site not being target of plasmin cleavage (Supplementary Fig. 7d, e). The latter was further confirmed when rather than entire muFasL ECD

(103–281), shorter muFasL ECD (134–281) was engineered into IgG1-muFasL[S] and tested (Fig. 4c-e, Supplementary Fig. 7d-e). No cleavage fragment was released by IgG1-muFasL[S] (Fig. 4f, Supplementary Fig. 7e) indicating that cleavage site being between 103 and 133 rather than at [143]PRSV[146]. On further mutational analysis, we discovered that [123]KV[124] in muFasL is the target of plasmin cleavage, as KV-AA substitution inhibited the cleavage (Fig. 4g). Considering that most MMP and ADAM10 cleavage sites (such [110]EL[111]/[113]EL[114]/[129]KQ[130]) are conserved in human and muFasL and are within stalk region, based on the data so far is it evident that selective plasmin cleavage of only huFasL at [144]RK[145] has the potential to release the shortest soluble fragment.

### Antibody-mediated interference of FasL plasmin cleavage FasL revives CAR-T bystander function

Previously described anti-FasL Nok1, Nok2 antibodies has been shown to pulldown MMP cleaved (134–281) soluble FasL[27]. Further Nok2 and its humanized version (Nok2h) antibodies recognize larger conformational epitopes on soluble FasL[28], hence, we hypothesized their possibility of interfering with plasmin cleavage of FasL. To test latter, we engineered and expressed Nok2 and Nok2h with murine IgG2a Fc (Supplementary Fig. 8a) using previously described sequences[28]. The Nok2 and Nok2h binding to FasL was confirmed using CD3/CD28 stimulated human PBMCs (Fig. 5a). When tested both Nok2 and Nok2h antibodies interfered with plasmin cleavage of huFasL and muFasL but not of IgG1 hinge cleavage by plasmin (Fig. 5b, c, Supplementary Fig. 8b, c). To further confirm these results, we made use of FasL stalk-region targeting antibody named 9F5 (sequence kindly provided by Dr. Patrick Legembre). 9F5 bound effectively on FasL overexpressing 1A12

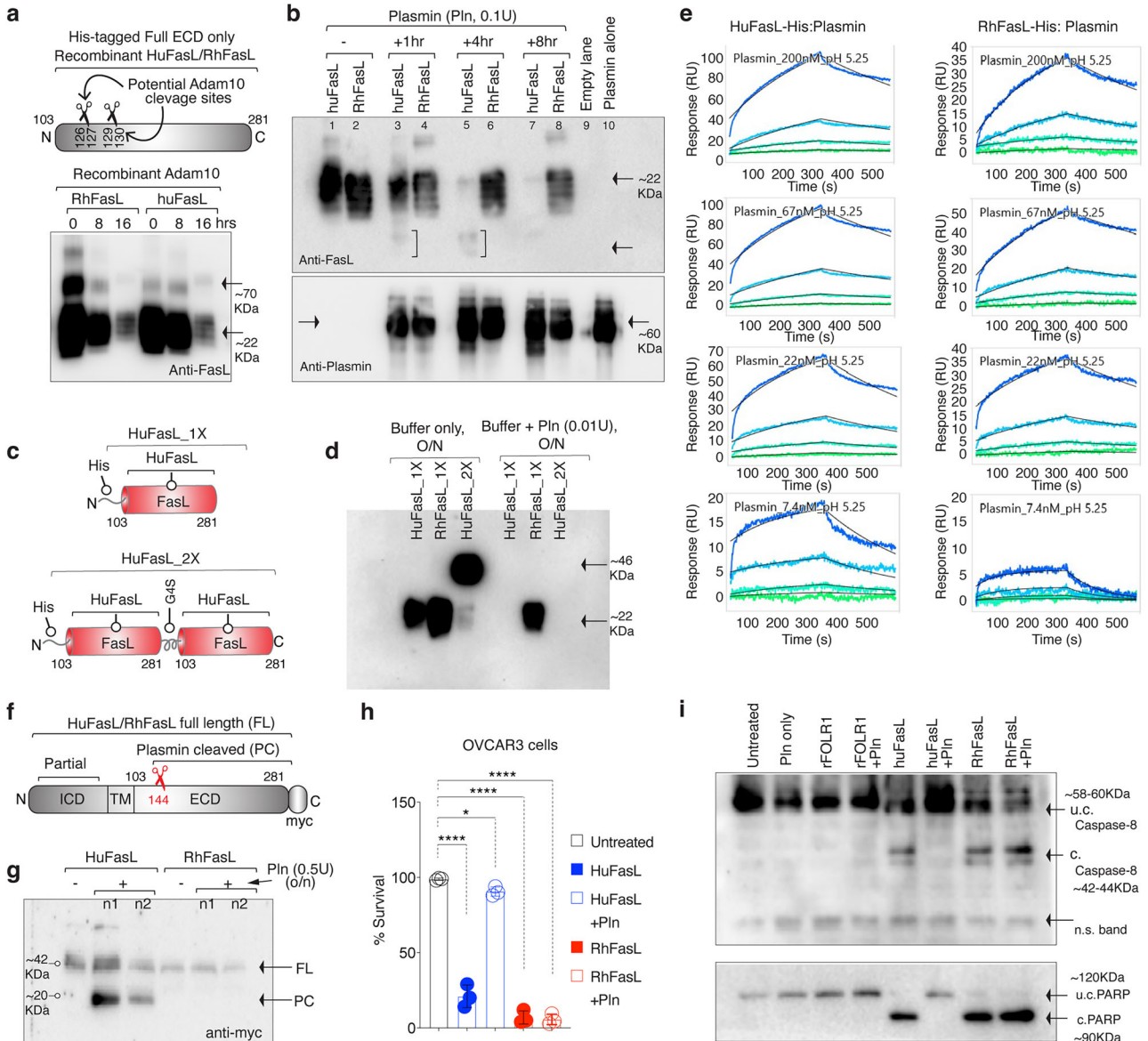

**Fig. 2 | HuFasL is preferentially cleaved by plasmin. a** Top: showing schematic and location of potential ADAM10 cleavage sites in huFasL. Bottom: RhFasL and huFasL were incubated with ADAM10 for indicated times followed by immunoblotting (reducing condition) with anti-FasL antibody. Representative blot is $n = 1$. **b** RhFasL and huFasL were incubated with plasmin (Pln) for indicated times followed by immunoblotting (reducing condition) for FasL and plasmin. Representative blot is $n = 1$, however it has been repeated multiple times in manuscript with additional parameters. **c** Schematic of genetic construction of his-tagged 1X and 2XFasL constructs described in (**d**). **d** Indicated RhFasL and huFasL were incubated with Pln for O/N followed by immunoblotting for FasL. Representative blot is $n = 1$. **e** The kinetic binding behavior of huFasL and RhFasL was analyzed against fixed and immobilized human plasmin at indicated pH values. **f** Schematic of myc-tagged membrane attached FasL ECDs. **g** Indicated membrane FasL-expressing cells were treated with Pln followed by immunoblotting against c-myc. Representative blot is $n = 1$. **h** Cell survival assay after HuFasL/RhFasL ± Pln treatment ($n = 3$, three independent experimental repeats) Untreated vs. huFasL, ****$p = <0.0001$; Untreated vs. huFasL+Pln, *$p = 0.0214$; Untreated vs. RhFasL, ****$p = <0.0001$; Untreated vs. RhFasL +Pln, ****$p = <0.0001 < 0.0001$; Unpaired, two-sided parametric t-test with no adjustments. **i** Same as (**h**), except lysates were analyzed for caspase-8 and PARP. Representative blot is $n = 1$, however it has been repeated in vivo with additional parameters, see Figs. 8, 9. Error bars in (**h**) are presented as mean ± standard deviation (SD). In (**i**) u.c. indicates uncleaved, c. indicates cleaved.

cells (Fig. 5d). We further engineered 9F5 into scFv (Supplementary Fig. 8d, e), which similar to Nok2h was effective in selectively blocking huFasL plasmin cleavage, but not IgG1 hinge cleavage by plasmin (Fig. 5e, f). Notably compared to Nok2h, 9F5 only partly inhibited to muFasL by plasmin (Fig. 5f). To understand the significance of larger FasL conformational epitopes engaging vs stalk region engaging specificities, we preincubated of huFasL either with Nok2 or Nok2h IgG2a or Nok2 or Nok2h scFvs or 9F5 scFvs, followed by addition of plasmin in-vitro. The mixture was added on ovarian tumor cells. Similar to previous data, plasmin pre-incubated IgG1-huFasL lost >90% activity, while RhFasL was effective ± plasmin. (Fig. 5g). Notably, preincubation

of Nok2h IgG2a (as compared to Nok2h scFv) with recombinant IgG1-huFasL interfered with its cell death (Fig. 5g, last two bars). Regardless, both Nok2h and 9F5 scFv's significantly rescued FasL killing activity (almost equivalent to aprotinin, a plasmin inhibitor, Pln-i) suggesting mechanistic blockade and interference with the plasmin accessibility of FasL[144]RK[145] region by the antibodies in scFv format (Fig. 5g).

FasL-Fas mediated off-target (bystander) killing of antigen negative tumor cells is critical for CAR-T activity against heterogenous tumors[8,10]. Thus, we tested if Nok2h and 9F5 scFv's would rescue CAR-T expressed FasL-mediated off-target cell death activation in presence of plasmin. Toward this end we made use of NaPi2b[29] (a cell surface

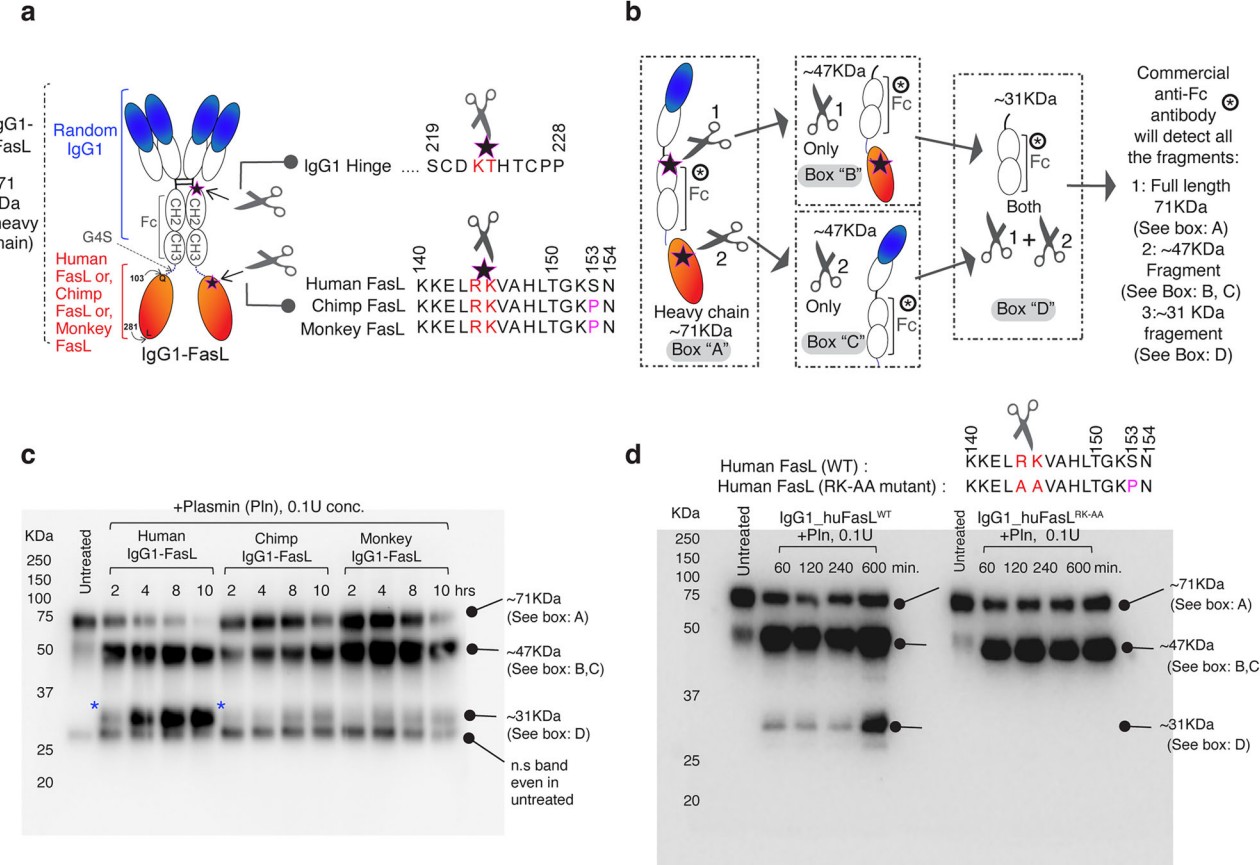

**Fig. 3 | HuFasL is selectively cleaved at $^{144}$RK$^{145}$ residues by plasmin. a** Schematic of engineering of random IgG1-conjugated FasL at the end of CH3 domain. Random IgG1 hinge was also engineered with an optimal Pln cleavage site. The detail sequences of two Pln cleavage sites are depicted by scissors and detailed sequences are shown. **b** If IgG1 hinge is cleaved by Pln: scissor-1, a 47 kDa fragment will release (see *box b*). If FasL is cleaved by Pln: scissor-2, another 47 kDa fragment will release (see *box c*). If cleavage occurs at both places (scissor-1 + 2), a ~ 31 kDa fragment will release (see *box d*). All cleaved fragments and full length (see *box a*) will have CH2-CH3 domain intact to be

recognized by anti-Fc antibody in reducing condition. **c** Time course Pln cleavage experiment as in (**b**). Only human IgG1-FasL produced the heightened 31 kDa band. A non-specific band of ~25 kDa is evident in all lanes including untreated. Representative blot is *n* = 1, however it has been repeated multiple times in manuscript with additional parameters (See Fig. 4, Supplementary Fig. 6). **d** Same as (**c**), except along with huFasL$^{144}$RK$^{145}$, huFasL$^{144}$AA$^{145}$ mutant was tested side-by-side. Representative blot is *n* = 2, see Supplementary Fig. 6c.

sodium-dependent phosphate transporter) negative HeyA8 cells[10]. We engineered NaPi2b scFv expressing CAR-T cells as published previously by us[10] (Fig. 5h) We consistently observed 15–20% CD8$^+$/G4S-linker$^+$ double positive CAR-T (Fig. 5i), which were highly effective against NaPi2b antigen positive OVCAR3 cells compared to NaPi2b antigen-negative HeyA8 cells as evident by granzyme-B activity (-10,000 pg/ml) (Fig. 5j). Next, we cocultured NaPi2b$^+$ OVCAR3 cells and NaPi2b$^-$ HeyA8 cells with CAR-T$^{NaPi2b}$ cells (2:3:1 ratio) in ±plasmin (Fig. 5k). In a different set random scFv controls and Nok2h and 9F5 scFv's were added in similar conditions. As expected, plasmin inhibited CAR-T expressed FasL-mediated caspase-8 activation (Fig. 5l). Importantly, both plasmin cleavage inhibiting Nok2h and 9F5 scFv's rescued it (Fig. 5l), confirming that plasmin-mediated FasL cleavage on T-cells would inhibit CAR-T cell-based immunotherapies in heterogenous conditions.

**Plasmin cleaved soluble huFasL does not self-aggregate into higher form**

Considering that the adam-10 and plasmin cleavage of huFasL will release soluble FasL fragments with partial or without trimerization motif (TRM), respectively (Fig. 6a), we cloned, expressed various soluble his-tagged FasL ECDs with different lengths. We named the full-length (103–281aa) ligand with entire trimerization domain, long FasL

[Long$^{(L)}$], post adam-10 cleavage (134–281aa) with partial trimerization domain, Shorter FasL [Short$^{(S)}$], post plasmin cleavage (146–281aa) with no trimerization domain, Shortest [Shortest $^{(ST)}$] (Fig. 6a). Similar to RhFasL$^L$, P153 harboring RhFasL$^S$ with partial TRM showed a higher trimer/monomer ratio than HuFasL$^S$ (Fig. 6b), further confirming the importance of CH/π interactions S153P. Above all, S153P substitution in both HuFasL$^L$ and HuFasL$^S$ normalized their trimer/monomer very similar to RhFasL$^L$ and RhFasL$^s$ (Figs. 1d, 6b). Strikingly even FasL$^{ST}$ form of human and rhesus FasL formed right size trimers (Supplementary Fig. 9a), confirming previous reports of stalk independent trimerization of FasL[30,31]. Notably, plasmin cleaved form of HuFasL$^{ST}$ showed >95% lower cell death capability against tumor cells (Fig. 6c) underscoring the negative impact of plasmin on FasL killing function.

Regardless of species, published studies unanimously agree that membrane FasL (FasL$^{Mem.}$) is highly cytotoxic. However, the same is not true for soluble FasL. For example, in murine knockout studies, the secreted soluble FasL lost cytotoxicity[32] compared to membrane FasL. In contrast, the secreted soluble form of human FasL, either the form of entire ECD (103–281aa) or cleaved in the middle of the stalk region (134–281aa), has been shown to be active in killing Fas$^+$ lymphocytes and other cells[33,34]. In addition, unlike huFasL, even differential sequence polymorphism of Fas ligand within mice strains has been described to have contrasting cytotoxic function of FasL$^{Mem. 35}$. In the

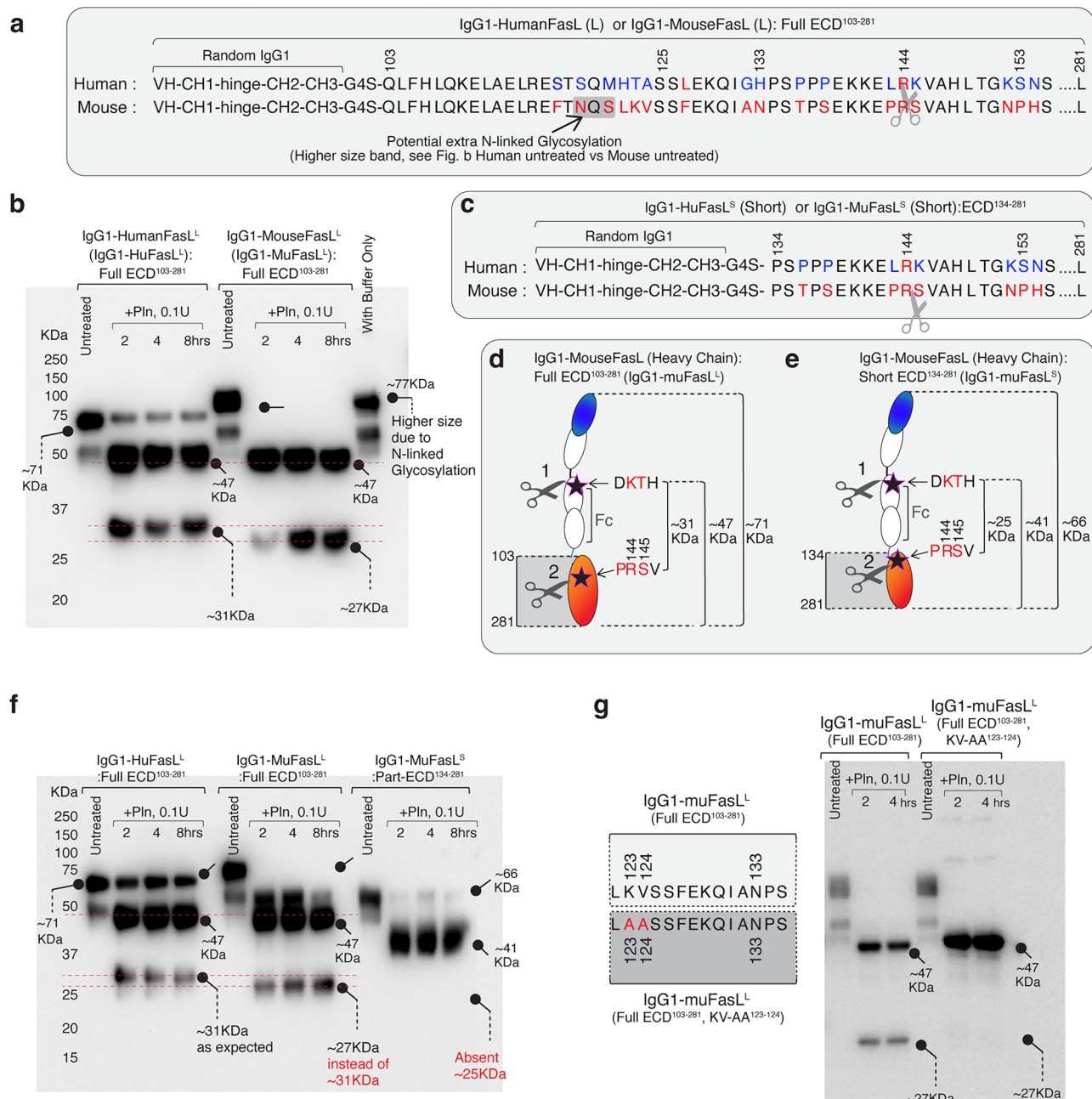

**Fig. 4 | Mouse FasL is not cleaved at** $^{144}$**RK**$^{145}$ **residues rather at site more proximal to transmembrane domain. a** Amino acid alignment of full length ectodomain of human (IgG1-HuFasL[L]) and mouse (IgG1-MuFasL[L]) FasL from Q103-L281 residues. An extra N-linked glycosylation site in mouse FasL is evident at $^{119}$NQS$^{121}$ while $^{144}$RS$^{145}$ substitution is present instead of $^{144}$RK$^{145}$. **b** Same in-vitro IgG1-FasL cleavage assay as in Fig. 3c, d except huFasL and muFasL were tested. Representative blot is $n=1$, however it has been repeated in manuscript with

additional parameters (See Fig. 4f). **c** Amino acid alignment of smaller human and mouse FasL (MuFasL[S]) ECD from P134-L281 residues with $^{44}$RS$^{145}$ still present. **d**, **e** Schematic and size comparison of IgG1-MuFasL[L] and IgG1-MuFasL[S]. **f** Same as (**b**), except IgG1-MuFasL[L] and IgG1-MuFasL[S] were compared side by side. Representative blot is $n=2$, see Supplementary Fig. 7e. **g** Same as (**f**), except along with WT IgG1-MuFasL[L], a $^{123}$KV-AA$^{124}$ mutant muFasL[L] was tested side-by-side. Representative blot is $n=1$.

context of tumor cells, we have also previously published the killing function of soluble HuFasL[L] against Jurkat and various solid tumor cell lines[10]. On the contrary, others have described either total or partial loss of cytotoxicity after processing of membrane FasL by MMPs[36,37]. When tested, unlike soluble huFasL, soluble muFasL was not an effective cell death activator (Fig. 6d). The latter supports previous studies[32] of loss of murine soluble FasL cytotoxic function. Notably, when mufasL was engineered with Nok2 and Nok2h antibodies in a bispecific configuration (Fig. 6e, Supplementary Fig. 9b), it was a highly

effective cell-death activator against tumor cells (Fig. 6f). As Nok2 and Nok2h specificity in Nok2h-huFasL and Nok2h-muFasL (Fig. 6e) have the potential to self-aggregate FasL via avidity due to multiple binding sites in a bispecific molecule, the data in Fig. 6d compared to Fig. 6f indicate potentially lower self-aggregation-function-of-soluble-muFasL-vs.-huFasL. To test this hypothesis that sequence differences in key residues of soluble human and murine FasL (Supplementary Fig. 7a) could differentially affect its higher-order clustering and killing activity, we added different forms (Fig. 6a) of human and murine FasL

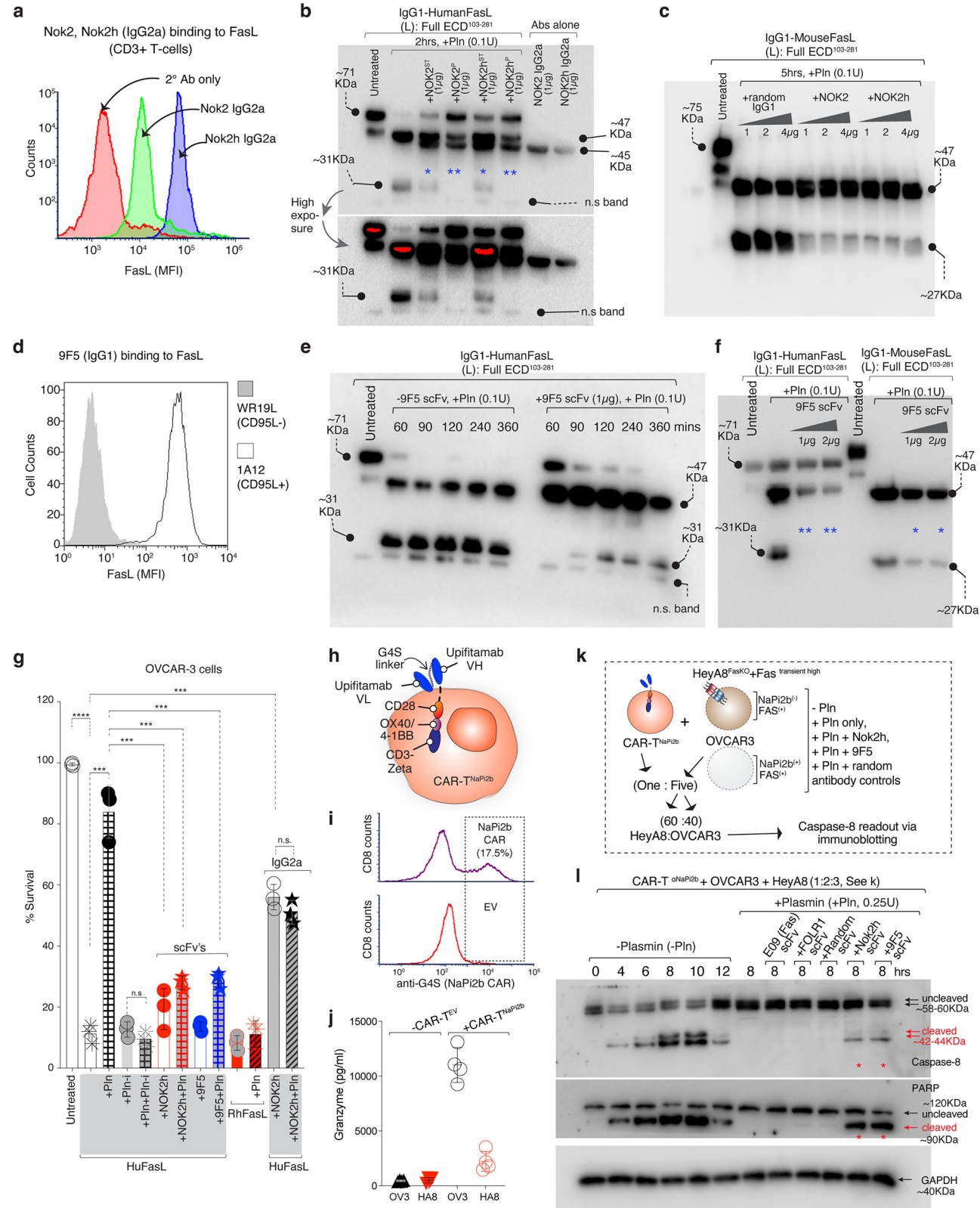

(FasL$^L$, FasL$^S$, FasL$^{ST}$) in the growing culture of murine 4T1 cells in serum-free media. 120 min later, the supernatant was precipitated, followed by direct loading onto non-reducing gels (with 60% lowered SDS conc.) to capture the potential aggregates. HuFasL$^L$ and HuFasL$^S$ formed aggregates, while various muFasL$^L$ and muFasL$^S$, although formed trimers, failed to form large aggregates (Fig. 6g). Same results were seen with human OVCAR3 cells (Supplementary Fig. 9c). As

HuFasL$^{ST}$ is not effectively cytotoxic (Fig. 6c), it similarly failed to form larger aggregates (Fig. 6g, Supplementary Fig. 9c). These results collectively support the importance of FasL aggregation in inducing cell death. We repeated the experiments using Jurkat cell suspension culture with serum-containing culture media. Similar to 4T1 and OVCAR3 cells, the precipitate run without a reducing agent and without pre-heating, formed higher aggregates in non-reducing (with 60% lowered

**Fig. 5 | FasL stalk-targeting antibodies interfere with plasmin cleavage in vitro and in CAR-T bystander Fas assay. a** Flow cytometry analysis of Nok2 and Nok2h IgG2a binding on CD3 positive cells. **b** Same in-vitro IgG1-huFasL cleavage assay as in Figs. 3, 4 except huFasL was preincubated with Nok2 and Nok2h IgG2a antibodies (37 °C) either 1 h prior (P) or at the same time (ST) of adding plasmin (Pln). Representative data in blot is $n = 3$ with experiments done with additional parameters, see Supplementary Fig. 8b, c. **c** Same as (**b**), except IgG1-MuFasL$^L$ was used. Representative blot is $n = 1$. **d** Flow cytometry analysis of 9F5 IgG1 against WR19L (huFasL$^-$) and 1A12 (huFasL$^+$) cells. **e** IgG1-huFasL$^L$ cleavage assay except 9F5 scFv was preincubated 1 h prior to adding Pln (37 °C). Representative blot is $n = 1$, however it has been repeated in vivo with additional parameters, see Fig. 8g. **f** Same as (**e**), except IgG1-MuFasL$^L$ was also compared with 9F5 scFv. Representative blot is $n = 1$, however it has been repeated in vivo with additional parameters, see Fig. 8g. **g** Cell survival analysis of OVCAR3 cells 24 hr post treatment with huFasL$^L$ (his-tag) pre-incubated with ± Nok2h or 9F5 scFv's or Nok2h IgG2a in presence of Pln. ($n = 3$, three independent experimental repeats of cell survival) Untreated vs huFasL, ****$p = <0.0001$; huFasL vs. huFasL+Pln, ***$p = 0.0002$; huFasL+ Pln-i alone vs huFasL

+Pln + Pln-i, nsp=0.1970; huFasL+Pln vs huFasL+Nok2h scFv, ***$p = 0.0005$; huFasL+Pln vs huFasL+Pln +Nok2h scFv, ***$p = 0.0004$; huFasL+Pln vs huFasL+Pln +9F5 scFv, ***$p = 0.0005$; RhFasL vs RhFasL+Pln, nsp=0.3641, huFasL vs huFasL+Nok2h IgG2a, ***$p = 0.0001$; huFasL+ Nok2h IgG2a vs huFasL+Pln+ Nok2h IgG2a, nsp =0.2059. Unpaired, two-sided parametric t-test with no adjustments. **h** Cartoon schematic of anti-NaPi2b (Upifitamab) scFv expressing CAR-T. **i** Flow cytometry of cell surface expression of CD8 and NaPi2b scFv (top), empty vector (bottom) CAR, as detected using anti-G4S linker antibody. **j** OVCAR3 (OV3) or HeyA8 (HA8) cells were co-cultured either with CAR-T$^{EV}$ or CAR-T$^{NaPi2b}$ cells at an effector/target (E/T) ratio 5:1 for 4 h followed by granzyme B measurement using ELISA ($n = 3+$ for granzyme B value measurements in three plus independent experimental repeats). **k** Schematic of CAR-T$^{NaPi2b}$ cells co-cultured with the 60:40 mix of OVCAR3 and Hey-A8$^{FasKO}$ overexpressing exogenous Fas in presence of ± Pln plus indicated scFv's on right. **l** Same as (**k**), except lysates were analyzed for caspase-8 and PARP via immunoblotting. Representative blot is $n = 1$. Error bars in (**g**) are presented as mean ± standard deviation (SD). In (**l**) u.c. indicates uncleaved, c. indicates cleaved.

---

SDS conc.) gels (Fig. 6h). Addition of DTT to the heated samples before loading eliminated higher aggregates (Fig. 6h, lane 1, 2, 5). Strikingly the HuFasL$^{ST}$ in both scenarios lacked proper aggregate formation (Fig. 6h, lane 5 vs 6).

The FasL aggregation is key for its function, and one well-studied inborn error of autoimmune lymphoproliferative syndrome (ALPS) centers around FasL$^{A274E}$ mutation[38]. Indeed, FasL$^{A274E}$ or FasL$^{A274D}$ failed to properly trimerize and effectively cluster (Fig. 6i-k, l: lane 1, 2). As the presence of a shorter side chain gly$^{246}$ and ala$^{247}$ next to the hydrophobic tyr$^{244}$ and leu$^{245}$ in FasL "G" beta-strand (Fig. 6i, j) is critical for interlocking FasL monomers together in trimer (Fig. 6k), we investigated potential sequence differences in mouse vs human FasL on "G" beta-strand region (Fig. 6j). Interestingly a critical R-H substitution is evident at residue 241. When tested, FasL$^{R241H}$ had a slightly different trimer profile but clustered similarly to WT (Fig. 6l, lane 4). Same was the case for *Gld* (F275L) mutation[31,39] which is known to cause lymphoproliferative disease similar to ALPS. (Fig. 6l, lane 5). Strikingly, the conserved sequence between R241 and A247 (SSYLGA) contains the YLGA motif, which is known to directly interact with the YLGA/FLGA motif of mouse and human c-met, respectively[40]. The direct steric binding of the c-met is potentially one of many possible mechanisms to interfere with proper FasL-mediated Fas aggregation and apoptosis in hepatocytes[40] and endothelial cells[41]. When tested, YLGA-AAAA mut FasL failed to trimerize and aggregate (Fig. 6l, lane 6). Regarding c-met expression, we observed a trend of low c-met expression and higher FasL sensitivity (Fig. 6m). Among tested tumor cells, the MDA-MB-468 cells expressing the highest c-met levels were fully resistant to soluble FasL (Fig. 6o) compared to OVCAR3 and MDA-MD-231.

Since endothelial HbMEC cells have been shown to be sensitive to plasmin-cleaved FasL, we tested their c-met expression. Notably, a significantly higher (>6-fold) c-met protein profile was evident in hBMEC cells (vs. OVCAR3), regardless of confluency (Fig. 6n). Next, we transiently transfected CHO-K cells with a pBOS-HFLD19 vector (Supplementary Fig. 10a, kindly provided by Dr. Shigekazu Nagata) that lacks proline-rich 8–69 cytoplasmic amino acids but has been used by numerous studies[42,43] to study membrane FasL (FasL$^{Mem.}$) mediated killing (See also Fig. 7h). The transient FasL$^{Mem.}$ expressing CHO-K cells were co-cultured with either low c-met expressing OVCAR3 or high c-met expressing HbMEC endothelial cells (Supplementary Fig. 10b). As expected, HbMEC cells were resistant to FasL$^{Mem.}$, with only partial sensitivity to soluble forms, including HuFasL$^S$ and HuFasL$^{ST}$ (Supplementary Fig. 10c). In contrast, HuFasL$^{ST}$ was ineffective against tumor cells (Fig. 6c, Supplementary Fig. 10c). When analyzed for cellular Fas and FADD clustering via running the total lysates in non-reducing gels (with 60% lowered SDS conc.), followed by Fas immunoblotting as published earlier[10,44], an opposite Fas and FADD clustering was evident in OVCAR-3 vs HbMEC cells. In OVCAR-3 cells, both FasL$^{Mem.}$ and FasL$^S$

clustered Fas and FADD (Supplementary Fig. 10d, e), while in HbMEC cells, both soluble FasL$^S$ and FasL$^{ST}$ partly clustered Fas and FADD (although at much lower levels than tumor cells). Curiously, FasL$^{ST}$ (plasmin cleaved) resulted in a different Fas oligomerization pattern in endothelial cells, as evident with slightly shorter Fas and FADD clusters (Supplementary Fig. 10d, e). Correspondingly, we observed differential caspase-8 activation and caspase-3 cleavage (Supplementary Fig. 10f). To confirm c-met mediated negative regulation of FasL$^{Mem.}$ is due to c-met inhibitory interactions with membrane FasL[40], we treated HbMEC cells with c-met neutralizing antibody named MetMab that imposingly enhanced HbMEC (but not OVCAR-3) killing by FasL$^{Mem.}$ (Supplementary Fig. 10c). The MetMab did not enhance HbMEC killing by plasmin cleaved soluble FasL form, potentially indicating differential FasL regulation upon cleaved by proteases. These results suggest that not only the species-specific sequence FasL polymorphism[35] that could affect its higher order aggregation capability but also differentially expressed surface regulators in a particular cell type or tissue, contributing to its differential cytotoxic outcome by membrane and soluble form[22]. Regardless, FasL higher-order aggregation is key for its cytotoxic function.

## Limited cytotoxicity of HuFasL against plasmin expressing tumor cells

Next, we tested tumor lines' ability to generate soluble FasL by analyzing their plasminogen-plasmin activation pathway protein levels. Unlike MDA-MB231 and HeyA8, colo-205 cells expressed significantly high uPAR/plasmin/uPA proteins with an expected size similar to primary endothelial HbMEC cells[22] (Fig. 7a). Interestingly, colo-205 cells were significantly less sensitive to huFasL compared to MDA-MB231 (Fig. 7b). The xenografted colo-205 tumors also expressed high uPAR and plasmin (Supplementary Fig. 11a) and were differentially sensitive to HuFasL (vs RhFasL) compared to MDA-MB231 tumors (Fig. 7c, d). As expected intertumoral injection of colo-205 tumors with plasmin inhibitor aprotinin resorted sensitivity to huFasL(Supplementary Fig. 11b).

Historically, plasminogen-system degrades stroma-derived fibrin in ovarian cancer (OvCa) ascites[45] and high levels of plasminogen-system inhibitor PAI-1 in ascites prolong disease free survival in late stage 3 OvCa patients[46,47]. Hence, to test in a clinically relevant scenario, we screened multiple OvCa patient-derived cell lines (PDC) and found a few chemoresistant PDC expressing elevated plasmin/uPA/uPAR levels (Fig. 7e). Similar to colo-205, OvCa PDC$^{1030}$ and PDC$^{1031}$ cells were significantly less sensitive to huFasL than RhFasL (Fig. 7f). Since lipid rafts are one of the key membrane domains for uPA function to convert plasminogen into plasmin[48,49], we treated HeyA8 and PDC$^{1030}$ and PDC$^{1031}$ cells with previously described uPA agonist A8 antibody[50] and carried out sucrose gradient sedimentation to separate caveolin

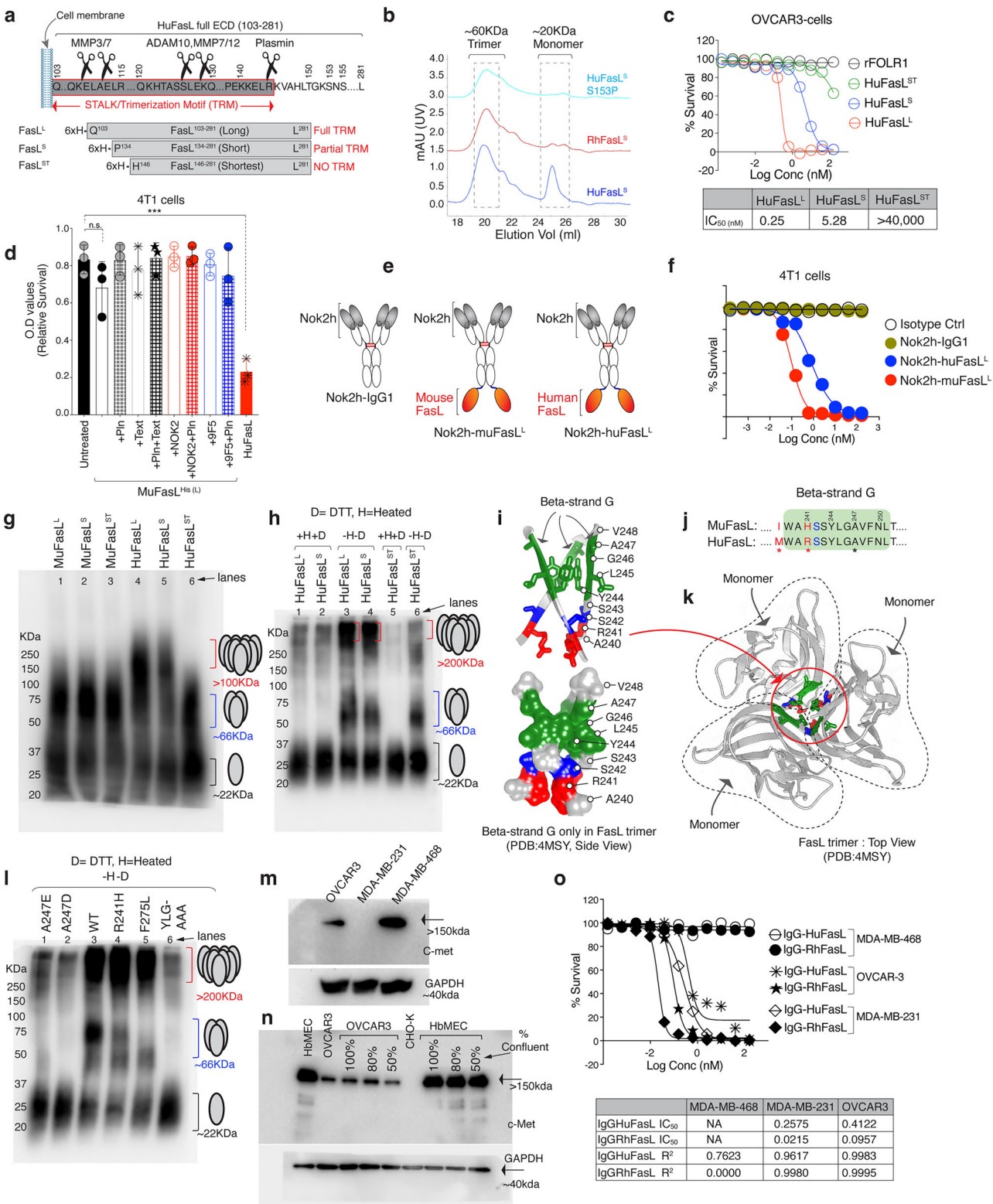

and cholesterol enriched lipid rafts microdomains from cadherin-family enriched non-rafts fractions (Supplementary Fig. 11c). Caveolin-rich fractions (fraction 2 and 3) were next run on SDS-PAGE followed by immunoblotting with plasmin. As expected, plasminogen was present in lipid rafts in both PDC[1030] and PDC[1031] cells but not in Hey-A8 cells (Supplementary Fig. 11d). More importantly, conversion of plasmino-gen to plasmin was evident in both PDC[1030] and PDC[1031] cells upon addition of uPA agonist antibody (Supplementary Fig. 11d). To further confirm plasmin activity in PDC[1031] and colo-205 cells, we made use of a

commercial colorimetric plasmin assay kit. Both PDC[1031] and colo-205 cells showed varied degree of plasmin function which was enhanced by uPA agonist A8 antibody (Fig. 7g). Similar to raft fractions, no plasmin activity was seen in Hey-A8 cells (Fig. 7g). To confirm membrane FasL plasmin cleavage by PDC[1031] cells, we transiently transfected CHO-K cells with pBOS-HFLD19 vector to express surface expressed FasL (Fig. 7h). Next, ovarian PDC[1031] cells were mixed and incubated at 37 °C with transient membrane huFasL (HuFasL[Mem.]) expressing CHO-K cells. The mixture was analyzed for surface FasL on CHO-K by flow

**Fig. 6 | Differential aggregation profile of human and murine soluble FasL.**
**a** Depiction of HuFasL$^{L(103-281)}$, HuFasL$^{S(134-281)}$ and plasmin-cleaved HuFasL$^{ST(146-281)}$ generation. **b** SEC chromatograms of indicated FasL$^S$ forms shown with different colors. huFasL$^{ST}$ and RhFasL$^{ST}$ are shown in Supplemental Figure 9a. **c** Cell survival assay after indicated HuFasL treatment. **d** Cell survival analysis of murine 4T1 cells 24 hr post treatment with his-tagged muFasL$^L$ pre-incubated with ± Nok2h or 9F5 scFv's ± Pln. ($n = 3$, three independent experimental repeats) Untreated vs. RhFasL, nsp= 0.1753; Untreated vs. huFasL, ***$p$ = 0.0005; Unpaired, two-sided parametric t-test with no adjustments. **e** Schematic of generation of bispecific Nok2h-FasL proteins used in (**f**). **f** Cell viability analysis of murine 4T1 cells treated either with Nok2h-IgG1 alone or Nok2h-IgG1-muFasL$^L$ and Nok2h-IgG1-huFasL$^L$. **g** Indicated various forms of his-tagged human and murine FasL were added onto 4T1 cells for 2 hr in serum free media followed by precipitation of supernatant. The samples were loaded (without boiling) on to non-reducing gels (with 60% lowered SDS conc.) to capture the potential aggregates by FasL immunoblotting. Representative data in blot is $n = 1$ with experiments done with additional parameters in

Fig. 6h, I and Supplementary Fig. 9c. **h** Indicated various forms of his-tagged human FasL were added into Jurkat cell suspension for 2hrs in serum containing media followed by precipitation of supernatant. The samples were loaded ±pre-heating or ±DTT on to non-reducing gels (with 60% lowered SDS conc.) followed by immunoblotting with anti-FasL antibody. Representative data in blot is $n = 1$. **i** Ribbon trace (top) and space filling (bottom) backbone structure of FasL G beta-strand in trimer. Green sticks from Y244-A247 represent c-met interacting YLGA motif. Red and blue sticks highlight R241 and S242. **j** Comparison of huFasL and muFasL G beta-strand sequence. **k** A ribbon trace structure of FasL trimer. The dotted circles show each of the monomer. The interface of monomers (specifically G beta-strand) is circled. **l** Similar to (**h**) various indicated huFasL mutants were analyzed for their trimerization and aggregation capability on Jurkat cells. Representative data in blot is $n = 1$. **m**, **n** Immunoblotting analysis of total c-met and GAPDH from indicated cell lysates. Representative data in blot is $n = 1$. **o** Cell survival analysis (post 36 hr) of indicated c-met expressing cells against increasing concentration of IgG1-huFasL or IgG1-RhFasL. Error bars in (**d**) are presented as mean ± standard deviation (SD).

cytometry (Fig. 7i, Supplementary Fig. 11e). For the control experiment, the HuFasL$^{Mem.}$ expressing CHO-K cells were mixed with PDC$^{1031}$ cells on ice just before flow analysis. As expected, a decrease in membrane FasL (from 40.5% to 15.5%) was evident within hours of incubation (Fig. 7i: case 2 vs case 3).

Next, we added IgG1-HuFasL on highly confluent PDC$^{1031}$ cells. After 2 hrs, supernatant was concentrated using ammonium sulfate followed by immunodetection of IgG1-HuFasL using anti-Fas and anti-Fc antibodies form the same samples (Fig. 7j). As the hinge of IgG1-HuFasL did not contain plasmin cleavage site (for this experimental design), we observed loss of 71kDa signal in anti-FasL immunoblot, while a 47kDa fragment was evident in anti-Fc immunoblot (Fig. 7k). These results further support FasL processing by endogenous cellular plasmin. Notably, similar to in vitro experiments (Fig. 5b, e), pre-incubation with both Nok2h and 9F5 scFv's rescued the cellular plasmin processing of FasL (Fig. 7k, lane 3 vs 4, 5, 7), similar to blockade of in-vitro processing. The leftover cells of same samples (-supernatant) showed partial caspase-8 and PAPR activation in corresponding lanes (Supplementary Fig. 11f). Further, in support, the grafted tumors of chemoresistant PDC$^{1031}$ did not respond to huFasL in both His-tag (Fig. 8a) and IgG1-FasL format (Fig. 8b). Still, RhFasL, Chimpanzee FasL (ChiFasL), and HuFasL$^{S153P}$ were effective (Fig. 8a, b). Moreover, intertumoral injection of plasmin inhibitor aprotinin rescued tumor cytotoxicity (Fig. 8b). Also, similar to interfering with plasmin-cleavage function, injection of Nok2h and 9F5 scFvs largely rescued antitumor function of huFasL against PDC$^{1031}$ tumors (Fig. 8c).

### Plasmin activity downregulates surface FasL expression on tumor infiltrated T-cells

To test if plasmin cleavage of human FasL could interfere with its killing function on activated T-cells in tumors, we tested three different mouse tumor cell lines (MC38, 4T1, ID8$^{OVA}$) to generate human FasL-expressing tumors for syngeneic studies. We consistently observed that huFasL transfected MC38, 4T1 tumor cells underwent cell death in the culture immediately upon the start of the selection. On the contrary, ID8$^{OVA}$ cells had difficulty expressing huFasL despite repeating these transfection experiments over 6 times, which led us to conclude that either huFasL mRNA or huFasL protein were toxic to cells similar to previously described FasL mRNA[51,52]. Regardless, when tested both 4T1 and MC38 murine cell lines (but not ID8$^{OVA}$, a murine ovarian cancer epithelial line) were sensitive to soluble huFasL protein (Fig. 8d). If the insensitivity of ID8$^{OVA}$ against huFasL could be due to high expression of plasmin/uPAR/uPA components, we confirmed their expression (Fig. 8e). Indeed, similar to patient-derived OvCa cells (PDC$^{1031}$), the processing and cleavage of huFasL was evident upon addition to the culture (Supplementary Fig. 12a, b). To test if the latter would also happen in tumors, we subcutaneously grafted ID8$^{OVA}$ tumors on their matched immune-competent hosts C57BL/6 animals

followed by intratumoral injection of huFasL (Supplementary Fig. 12c). Harvested tumors were analyzed for caspase-8 cleavage, an indicator for FasL function. No caspase-8 activation was evident (Supplementary Fig. 12d, $n = 2$). Strikingly intertumoral injection of aprotinin and plasmin cleavage inhibiting antibodies restored caspase-8 function (Supplementary Fig. 12d). As aprotinin is not a highly selective plasmin inhibitor and could also partly interfere with other proteases, we made use of a pan ADAM10, ADAM17 inhibitor named GI-254023X, which was ineffective (Supplementary Fig. 12d), further supporting plasmin-mediated regulation of FasL in ID8$^{OVA}$ tumors. Although both Nok2h-IgG2a and 9F5 scFv were equally effective in-vitro (Fig. 5), Nok2h-scFv showed higher caspase-8 function reversal in tumors (Fig. 8f, g), hence we used the latter for additional murine studies. As shown earlier (Fig. 4), although at residues different than humans ($^{123}$KV$^{124}$ vs $^{144}$RK$^{145}$), muFasL is also a direct target of plasmin cleavage, hence, we hypothesized cleavage of membrane FasL on tumor infiltered murine T-cells in grafted ID8$^{OVA}$ tumors. To confirm plasmin mediated loss of T-cell membrane FasL in the context of tumor-immune interactions, we subcutaneously grafted uPA/uPAR/plasmin component negative MC38 tumors (Fig. 8h) and plasmin protease component positive ID8$^{OVA}$ tumors on their matched immune-competent hosts C57BL/6 animals (Fig. 8h). As published previously by us[53] and others[54,55], both MC38 and ID8$^{OVA}$ tumors express PD-L1 and respond to avelumab antibody which is both human and mouse cross-reactive[54,55]. The randomly selected animals from both tumor-type groups bearing ~75–100 mm$^3$ tumors were injected intraperitoneally (i.p.) every third day with a 50-µg dose of avelumab IgG4, avelumab ± Nok2h scFv. We and others have previously described activation and tumor mobilization of T-cells-mediated PD-L1+ syngeneic tumor regression by avelumab[53,54]. Next to test the plasmin mediated potential cleavage of muFasL on murine tumor-infiltrating T lymphocytes, we harvested and enriched CD3+ population from the size matched tumors of various treatment as published[53,55]. The tumor isolated and CD3 gated T-lymphocytes were analyzed for membrane FasL in both sets (Supplementary Fig. 13a). We observed a significantly higher percentage (>10-fold) of FasL$^+$/CD3$^+$ double positive cells in anti-PD-L1 treated (vs isotype control) in MC38 tumors (Fig. 8i). Strikingly, the FasL$^+$/CD3$^+$ double positive cells were only two-fold higher in in anti-PD-L1 treated (vs isotype control) ID8$^{OVA}$ tumors (Fig. 8j). Notably, the injection of 50 µg dose of FasL plasmin-cleavage interfering Nok2h-scFv (Fig. 5) significantly restored (almost >5-fold) FasL$^+$CD3$^+$ population in ID8$^{OVA}$ tumors but not in MC38 tumors (Fig. 8k, Supplementary Fig. 13b, c, $n = 3$). These results further support membrane FasL regulation by plasmin on tumor infiltrated and activated T-cells.

Next to test human FasL cleavage in the tumor-immune context, we intraperitoneally injected ID8$^{OVA}$ tumor harboring C57BL/6 animals either with 50 µg avelumab alone or 50 µg avelumab + 25 µg huIgG1-FasL. In third set 50 µg avelumab + 25 µg huFasL + 50 µg Nok2h-scFv

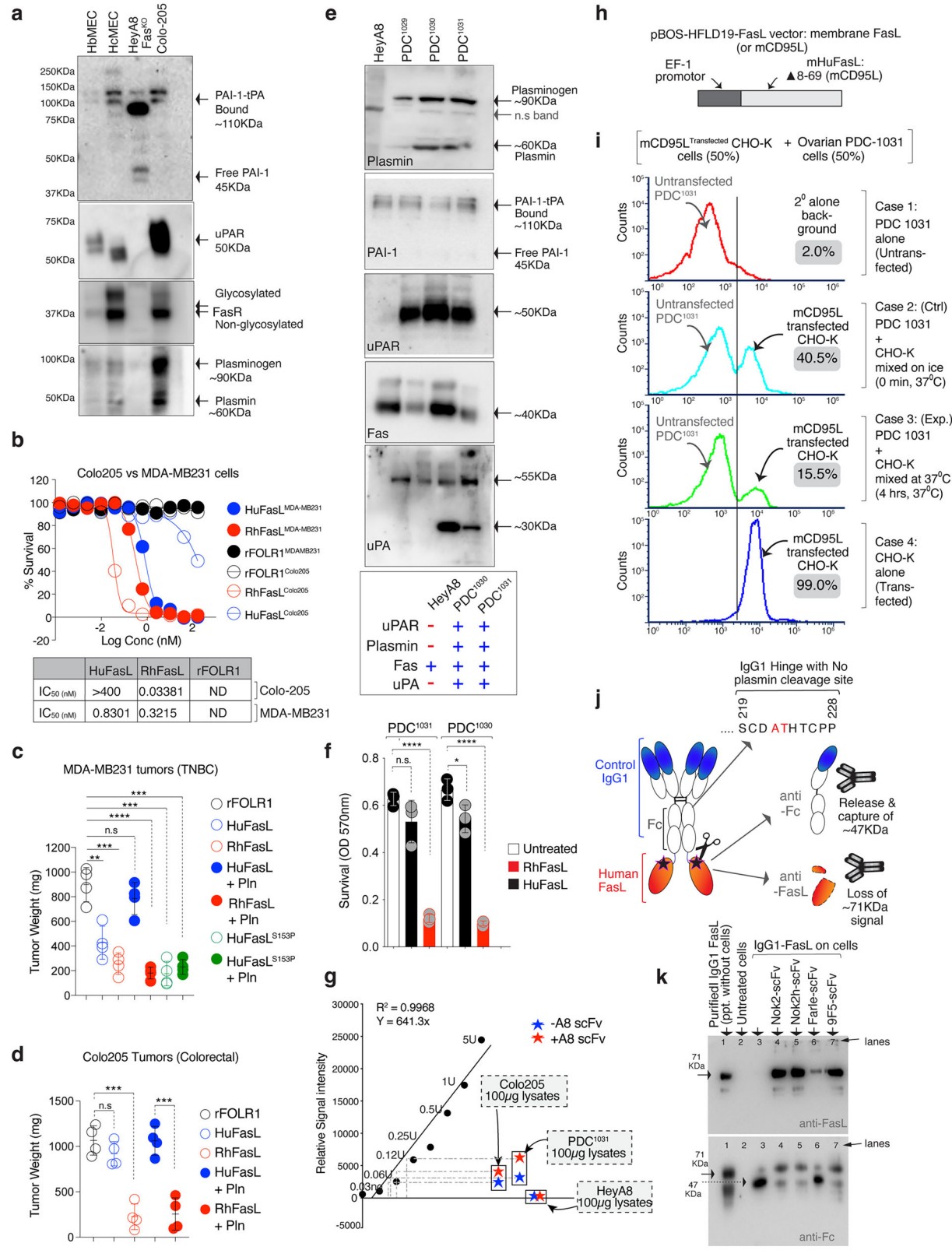

were injected along with the isotype control (Fig. 9a). Harvested tumors after five doses were analyzed for intracellular IFN-γ expression (a key activation marker for cytotoxic T-cell activity) in the enriched CD8 population using flow cytometry studies. All avelumab treatment groups (regardless of FasL co-injections) showed almost similarly elevated IFN-γ expression in CD8[+] cells (Fig. 9b, c, Supplementary Fig. 13d). We next analyzed the leftover tumor lysates for caspase-8

activity. Strikingly, caspase-8 activation was selectively evident in avelumab +huFasL+Nok2h-scFv treated tumors (Fig. 9d, $n = 3$). Along with higher CD8 signal, both granzyme and perforin levels were noticeably elevated in all avelumab treated tumors indicating T-cell activation (Fig. 9d). As granzyme is also known to cleave caspase-3 (in caspase-8 independent manner), the total caspase-3 levels were undetectable in all avelumab treated tumors (Fig. 9d). Notably, the

**Fig. 7 | Limited cytotoxicity of HuFasL against plasmin expressing cancer cells and tumors. a** Lysates from indicated cells on top were tested for PAI-1, uPAR, FasR, and plasmin expression. Representative data in blot is $n = 1$, however blot has been partly repeated with in vivo tumor parameter in Supplementary Fig. 11a. **b** Cell survival assays of indicated tumor cells with indicated treatments. **c** 6–8 week-old mice bearing SQ tumors of MDA-MB231 cells were i.p. injected with 50 µg of indicated FasL variant ± Pln every third ($n = 4$–6). After 4 weeks, isolated tumor weights were quantified ($n = 4$ mice) rFOLR1 vs. huFasL, **$p = 0.0028$; rFOLR1 vs. RhFasL, ***$p = 0.0002$; rFOLR1 vs. huFasL + Pln, nsp = 0.2775; rFOLR1 vs. RhFasL+ Pln, ****$p = <0.0001$; rFOLR1 vs. huFasL$^{S153P}$, ***$p = 0.0001$; rFOLR1 vs. huFasL$^{S153P}$+ Pln, ***$p = 0.0001$; Unpaired, two-sided parametric t-test with no adjustments. **d** 6–8 week-old mice bearing SQ tumors of colo205 cells were i.p. injected with 50 µg of indicated FasL varian (±Pln) every third ($n = 4$ mice). After 4 weeks, isolated tumor weights were quantified. ($n = 4$) rFOLR1 vs. huFasL, nsp = 0.2141; rFOLR1 vs. RhFasL, ***$p = 0.0002$; rFOLR1 vs. huFasL + Pln, nsp= 0.9753; huFasL + Pln vs. RhFasL+ Pln, ***$p = 0.0005$. Unpaired, two-sided parametric t-test with no adjustments. **e** Lysates from indicated patient-derived OvCa cells indicated on the top were tested for PAI-1, uPAR, FasR, plasmin and uPA expression. Representative data in blot is $n = 1$. **f** HuFasL and RhFasL treated patient-derived cells were subject to cell survival assay

after 48 h. ($n = 3$, three independent experimental repeats) PDC$^{1031}$: Untreated vs. huFasL, nsp = 0.1988; Untreated vs. RhFasL, ****$p = <0.0001$; PDC$^{1030}$: Untreated vs. huFasL, *$p = 0.0492$; Untreated vs. RhFasL, ****$p = <0.0001$; Unpaired, two-sided parametric t-test with no adjustments. **g** The normalized relative intensities of colorimetric signals were plotted for the increasing plasmin conc. (0.03-5U range) in presence of assay buffer and provided substrate (filled black circles) and against constant 100 µg membrane cellular lysates of indicated cell lines in presence (red star) or absence (blue star) of uPA agonist A8 scFv. **h–i** Ovarian PDC$^{1031}$ cells were mixed and incubated at 37 °C with transient membrane huFasL (mHuFasL lacking proline rich 8–69 cytoplasmic amino acids) expressing CHO-K cells under EF-1 promoter. The mixture was analyzed for surface FasL on CHO-K by flow cytometry. In control the mHuFasL+ CHO-K cells were mixed on ice just before flow analysis. **j** Schematic of IgG1-huFasL experiment used in (**k**). **k** IgG1-HuFasL was added on to confluent PDC$^{1031}$ cells. After 2 h, supernatant was concentrated using ammonium sulfate followed by immunoblotting using anti-Fas and anti-Fc antibodies form the same samples. Representative data in blot is $n = 1$, however the experiment and blot has been repeated with a different cell line. See Supplementary Fig. 12b. Error bars and data in (**c, d, f**) are presented as mean ± standard deviation (SD).

---

cleaved caspase-3 was highest evident in avelumab +huFasL+Nok2h-scFv treated tumors indicating huFasL and caspase-8 mediated gain of function (Fig. 9d). Furthermore, the huFasL and caspase-8 mediated gain of function reduced tumor burden significantly than anti-PD-L1 avelumab alone (Fig. 9e). These results at molecular and biochemical level support loss of FasL-mediated killing activity in plasmin-enriched tumors.

## Discussion

The down-regulation of immune-checkpoint receptors (PD1, CTLA4, TIGIT, etc.), up-regulation of cytokines (IFN-γ, TNF-α, IL2), and activation of granzyme and perforin, etc., remained the primary readouts to characterize the antitumor potential of tumor-infiltrated and activated T-cells[56]. In addition, an auxiliary tumor cell killing mechanism that heavily relies on T-cell (or CAR-T) expressed FasL-mediated extrinsic apoptotic cytotoxicity of Fas$^+$ cancer cells is critical in human clinical trials[8]. Using human, murine, and nonhuman primate systems, we have discovered an evolutionary regulatory mechanism of high clinical significance in dictating FasL-mediated auxiliary killing of tumor cells by activated T-cells.

Consistent with previous reports that have described proline to serine substitution contributing to increased protein susceptibility to proteolysis[57], FasL with S153 was a preferable plasmin substrate as compared to P153. We repeatedly found that huFasL plasmin cleavage due to P153S substitution was not limited to the membrane-expressed form; instead, every tested form, including soluble His-tag or Fc-Tag FasL, behaved similarly. Further, similar to the previously described endothelial cell migration context[22], exogenously added plasmin and endogenously tumor-enriched plasmin cleaved huFasL[22]. In terms of mechanism of action, we carried out the FasL saturation experiments using an array of previously described experimental FasL antibodies[28]. Strikingly, these antibodies (in small scFv format) selectively blocked plasmin cleavage while not significantly interfering with the FasL engagement of the Fas receptor, as evident with the effective tumor cell-killing function in vitro and against the grafted tumors in vivo. The latter is because antibody-mediated engagement of FasL structural epitopes likely limits plasmin accessibility to the cleavage site ($^{144}$RK$^{145}$) without interfering with the receptor binding and oligomerization capability[31]. We further confirmed these results with another scFv (named 9F5), which selectively engages the FasL stalk region near S153 residue. Nonetheless, if the P153-mediated reposition of $^{144}$RK$^{145}$ could also interfere with the optimal accessibility of the plasmin catalytic triad, RhFasL-plasmin structure studies could provide insights. Altogether, in light of protease upregulation being a key event in cancer metastasis[58], our findings systematically explain potentially

unfavorable antitumor outcomes of T-cell-based immunotherapies against plasmin overexpressing heterogenous solid tumors[59]. Moreover, these results imply that co-targeting of T-cell activating strategies with the plasmin inhibitors can potentially intensify the auxiliary FasL death receptor pathway in response to immunotherapies.

Besides plasmin cleavage, comparison of cell-killing, and self-aggregating function of various huFasL with muFasL soluble forms, our results agree with the published murine FasL knockout studies[32]. Regarding the cytotoxic function of soluble huFasL, there are some disagreements in the published literature. For example, there are multiple studies[22,34,37,60] that support effective cytotoxicity of recombinant soluble human FasL alone or in conjugation with peptides/antibodies[61,62] against various cell types, although the consistent theme in literature in various tissues (except endothelial tissue[22]) indicates the higher cytotoxicity associated with membrane-bound FasL. We hypothesized and proved earlier by using DR5 agonist antibodies[63] and others using hexavalent TRAIL[64] (which signals similar to FasL) and here using Nok2-FasL and Nok2h-FasL bispecific configurations (Fig. 6e, f) that the higher activity of membrane FasL is due to high-avidity driven clustering of Fas receptor by membrane-anchored FasL. The endothelial disparity presented in our data is supported by published studies indicating that c-met interacts not only with FasL[40] but also with Fas receptor to inhibit its function selectively in endothelial system[41]. Unlike plasmin-cleaved FasL-mediated cell-death[22], there are also published reports indicating either a significant[37] or complete loss[65] of its cell death function or cell death independent gain of calcium signaling function in promoting lymphocyte cell migration[25] by the protease cleaved soluble form of huFasL.

To understand some of these observed cell-death discrepancies, we generated additional previously described FasL mutants such as ALPS mutant (A247E)[38], *Gld* mutant (F275L)[31,39], and YLGA mutant[40]. Consistent with the literature, a mutation in the most distal c-terminal region of FasL (such as E271A[10] and *Gld* mutant F275L[31,39]) disrupts its apoptotic function (due to loss of binding to the receptor) but did not block its aggregation[31]. Similar to *Gld* mutation[31,39], any form of soluble FasL (FasL$^L$, FasL$^S$, etc.) generated with a His or Fc-tag next to its c-terminal was unfunctional in our hand, confirming the importance of the c-terminal in binding to Fas receptor. Furthermore, in agreement with published literature[40], the mutation in the YLGA motif located on the G-beta strand at the FasL trimer interface abolished its higher order aggregation, a key step for most TNF-α superfamily death receptors[60,66]. The YLGA motif of FasL interacts with c-met[38]. It is expected that the presence of a large receptor tyrosine kinase (RTKs) family member such as c-met on the

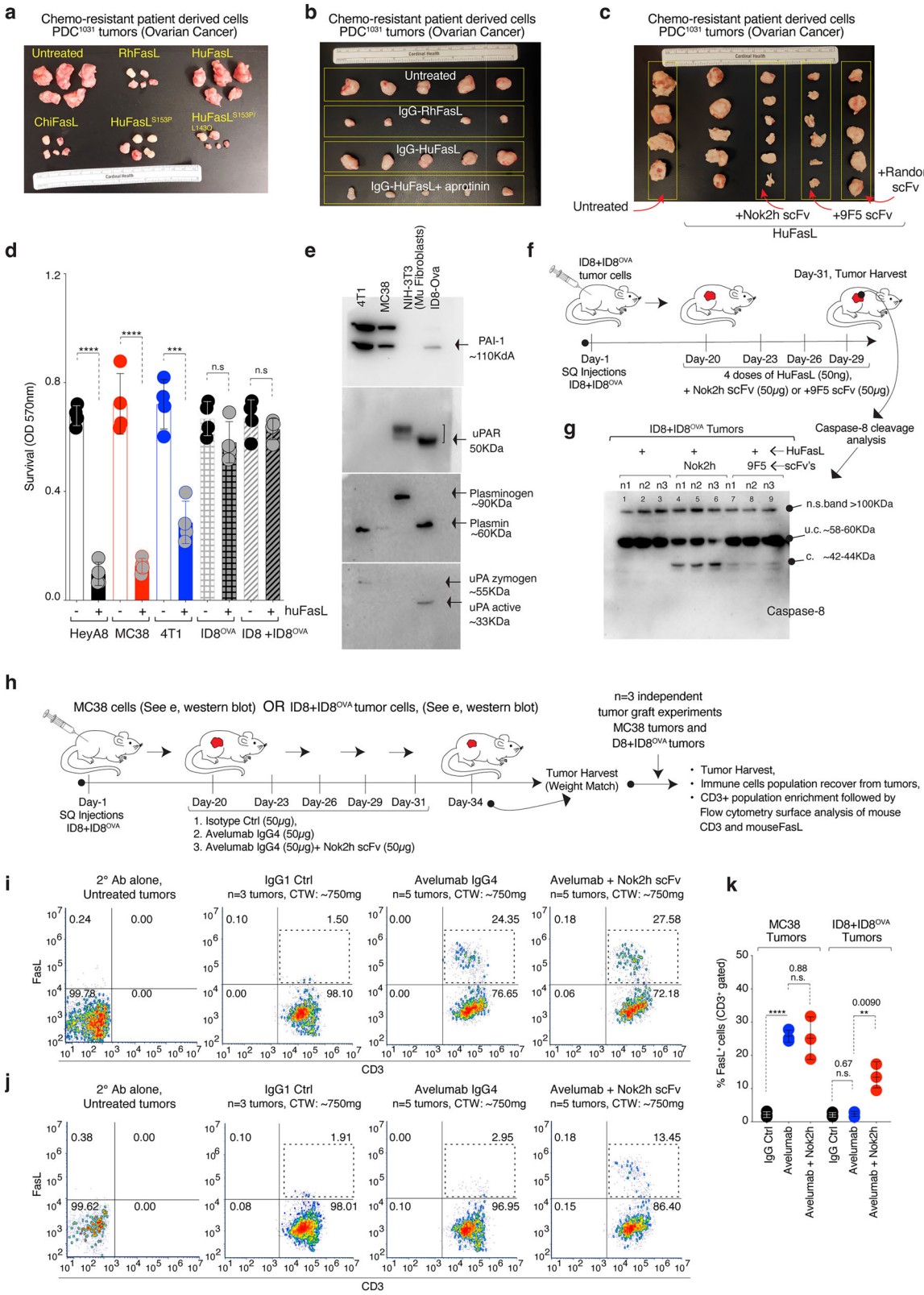

cell membrane close to FasL or Fas[41] either on the same cell or neighboring cell surface could interfere with the membrane FasL clustering function in tissues such as endothelial system[22]. Notably, similar steric interference with FasL resulted in inhibition of its cytotoxic function by large molecules (~150 kDa) such as Nok2 and Nok2h IgG2a but not by small size (~25 kDa) Nok2 and Nok2h scFv's despite targeting the same epitopes. Moreover, our results of

acquired FasL sensitivity in high c-met expressing cell lines with the c-met neutralizing clinical antibody (MetMab) further indicate that targeting the FasL aggregation steric inhibition could be partly responsible for the different outcomes observed by some studies[32,65]. For example, another published report has described DR5[67] as an interacting partner of FasL in human fibroblasts. Thus, it is highly plausible to hypothesize that unidentified surface proteins,

**Fig. 8 | Decreased membrane FasL expression on infiltrated T-cells in murine tumors with elevated plasmin function. a** 8-week-old mice bearing SQ tumors of OvCa PDC[1031] cells were i.p. injected with 50 μg of indicated his-tagged FasL variant every third day. After 4 weeks, isolated tumors from each group were imaged (n = 4–6). **b** Same as (**a**), except IgG1-FasL was used as indicated. **c** Same as (**a**), except IgG1-FasL was used alongside of indicated scFv's as indicated. **d** Cell survival assay using indicated murine tumor cell lines after IgG1-HuFasL treatment. (n = 3, n = 3, three independent experimental repeats) HeyA8: Untreated vs. huFasL, ****p = <0.0001; MC38: Untreated vs. huFasL, ****p = <0.0001; 4T1: Untreated vs. huFasL, ***p = 0.0003; ID8[OVA]: Untreated vs. huFasL, nsp = 0.1277; ID8 + ID8[OVA]: Untreated vs. huFasL, nsp = 0.3585; Unpaired, two-sided parametric t-test with no adjustments. **e** Lysates from indicated murine tumor cell lines on top were tested for PAI-1, uPAR, uPA, and plasmin expression. Representative data in blot is n = 1. **f, g** 6–8 week-old C57BL/6 mice were SQ grafted with ID8 + ID8[OVA] tumor cells. After 3 weeks (-20days), indicated treatments were given intraperitoneally every 3rd day for a total of five doses. Indicated Nok2h and 9F5 scFv were directly injected in tumors. At day 34–35, harvested tumor grafts (n = 3) were subjected to lysates. Following protein quantitation, lysates were analyzed for capsase-8 cleavage via immunoblotting. Representative data in blot is n = 1, however blot has been repeated with in vivo with additional parameters in Supplementary Fig. 12d. **h** Same as

(**f**), except two different set of experiments were carried out side-by-side using MC38 SQ tumor grafts (plasmin⁻) and ID8 + ID8[OVA] SQ tumor grafts as indicated. After 3 weeks (-20 days), animals were intraperitoneally (i.p.) injected either with 50 μg IgG control, avelumab (anti-PD-L1) or avelumab + Nok2h scFv every 3rd day for a total of five doses. At day 34–35 tumors were harvested from both MC38 and ID8 + ID8[OVA] sets. Next, total immune population were recovered from weight matched tumors using Ficoll-Paque density gradient centrifugation (pooled by treatment group, see methods) and the T-cell population was enriched using CD3 microbeads. The enriched CD3⁺ T-cells were analyzed for surface FasL expression using flow cytometry (Supplementary Fig. 13a for Gating strategy). **i, j** Recovered CD3⁺/FasL⁺ double positive populations from MC38 and ID8 + ID8[OVA] tumors. **k** Percentage of CD3⁺/FasL⁺ double positive cells from three independent experiments. (n = 3, independent biological replicates from weight matched tumors, 3–6 animal used for tumor recovery) MC38 tumors: IgG vs. Avelumab, ****p = <0.0001; MC38: Avelumab vs. Avelumab + Nok2h scFv, nsp= 0.8884; ID8[OVA]: IgG vs. Avelumab, nsp= 0.6748; ID8[OVA] tumors: Avelumab vs. Avelumab + Nok2h scFv, **p = 0.0090; Unpaired, two-sided parametric t-test with no adjustments. Error bars and data in (**d, k**) are presented as mean ± standard deviation (SD). In (**g**) u.c. indicates uncleaved, c. indicates cleaved.

regulators, or other steric mechanisms in a cell-type or tissue-type dependent context could modulate contrasting outcomes of FasL signaling regarding cell death and inflammation[25,68].

Besides the disparity in cancer development, brain size is one of the drastic physical attributes between human and other primates. The latter is reflected in the progressive cerebral cortex enlargement (-3 times larger) in humans[69] having twice as many cells in the cortex as the brain of a chimpanzee[70]. Furthermore, the molecular mechanism responsible for microcephaly (a condition with reduced brain cortex size) is due to an imbalance between cortical progenitor cell production and their programmed cell death[69,71]. Indeed, the key proteins associated with microcephaly regulate the cell cycle, mitotic spindle plus centrosome function, and cell death during brain development[72]. It has been described that the abnormal spindle-like microcephaly associated (ASPM) and microcephalin (MCPH1) genes, etc., have been undergoing robust adaptive evolution in the human lineage since splitting from chimpanzees. It is estimated that ASPM acquires positive progenitor cell growth promoting advantageous amino acid substitution in humans roughly every 300,000–400,000 years[73]. Because of the latter, overamplification of ASPM has been described in various human cancers[74], and loss of ASPM results in increased cell death of the cerebral cortex[75]. Similarly, in the ventricular zone of early developing embryonic mammalian brain, FasL is expressed on neural progenitor cells alongside Fas in the neighboring cells[76], where Fas-mediated programmed cell-death is critical during early forebrain development[77] and closure of the hindbrain neural tube[78]. Notably, high expression of plasminogen and tPA mRNA is evident in early mouse brain development (E10.5–E17.5), which coincides with a period of active migration of neuronal progenitor cells from the ventricular zone into the developing pons, medulla, and cerebellum[79]. Altogether, in light of human brain size enlargement, similar to the evolution of cell growth-enhancing primary microcephaly genes, hypothetically, the advantageous P153S amino acid substitution in huFasL to limit Fas-mediated programmed cell death of neural progenitor cells in the ventricular zone is foreseeable. As tradeoffs are a fundamental assumption in all evolving models[80], it is likely a probability that the evolutionary P153S substitution is an example of unfavorable concession on cell-death function concerning cancer in tPA/uPA and plasmin-rich microenvironment (Fig. 9f working model). As an example of evolving tradeoffs in terms of protein expression, in our >20 side-by-side protein expression studies (CHO-system), we have consistently observed increased overall expression of huFasL[WT] vs huFasL[S153P] (Supplementary Fig. 14). If the evolution-driven fine-tuning of higher human FasL expression was the additional driving factor for this acquired P153S

mutation in humans (over nonhuman primates) for a need-based additional layer of apoptosis regulation, the latter also could not be ruled out.

Further, suppose the heightened cancer susceptibility in modern humans is a consequence of evolutionary selection pressure for larger cognitive brain size. In that case, further investigations are needed. Regardless, similar to previously reported findings of p53[21], BRCA2[81], etc., our data with FasL support that more genetic variations in cancer pathways underwent positive selection in chimpanzee evolution[82], rendering higher cancer susceptibility in humans[83].

## Methods

All experiments were authorized to carry out under Biological Use Authorization (BUA) from the UC Davis Institutional Biosafety. All mouse experimental procedures followed the UC Davis Animal Care and Use Committee (IACUC) approved protocol and guidelines for the proper and humane use of animals in biomedical research were followed.

### Cloning of various recombinant FasL protein variants

All human, murine, rhesus recombinant FasL WT and mutants [either 103–281 amino acids: HuFasL[L], HuFasL-S153P[L] or 134–281 amino acids: HuFasL[S], HuFasL-S153P[S] or 146–281 amino acids: HuFasL[ST], HuFasL-S153P[ST]] either his-tagged or IgG1-Fc tagged etc. were cloned and expressed similar to HEK cells as published earlier[84]. The DNA sequences were retrieved from publicly available resources (NCBI or PDB database) and synthesized as gene string with N-terminal his-tag or C-terminal his-tag or IgG1-Fc tag etc. using Invitrogen GeneArt gene synthesis services. After PCR amplification, DNA was gel purified and inserted into the pCDNA 3.1 vector under the CMV promoter. We cloned it using In-Fusion HD Cloning Kits (Takara Bio). EcoR1 digested vector was incubated with overlapping PCR fragments of various different recombinant DNAs with infusion enzyme (1:5, vector: insert ratio) at 55 °C for 15 min, followed by an additional 10 min incubation on ice after adding *E. coli* StellarTM cells (Clontech). Transformation and bacterial screening were carried out using standard PCR with primers spanning EcoR1 sites. Confirmed bacterial colonies were Sanger sequencing upon PCR, followed by maxiprep and transfection of CHO cells.

### Recombinant FasL protein expression and purification

Freestyle CHO-S cells (Invitrogen) were cultured and maintained according to the supplier's recommendations (Life technologies) biologics using freestyle CHO expression system (life technologies) and as

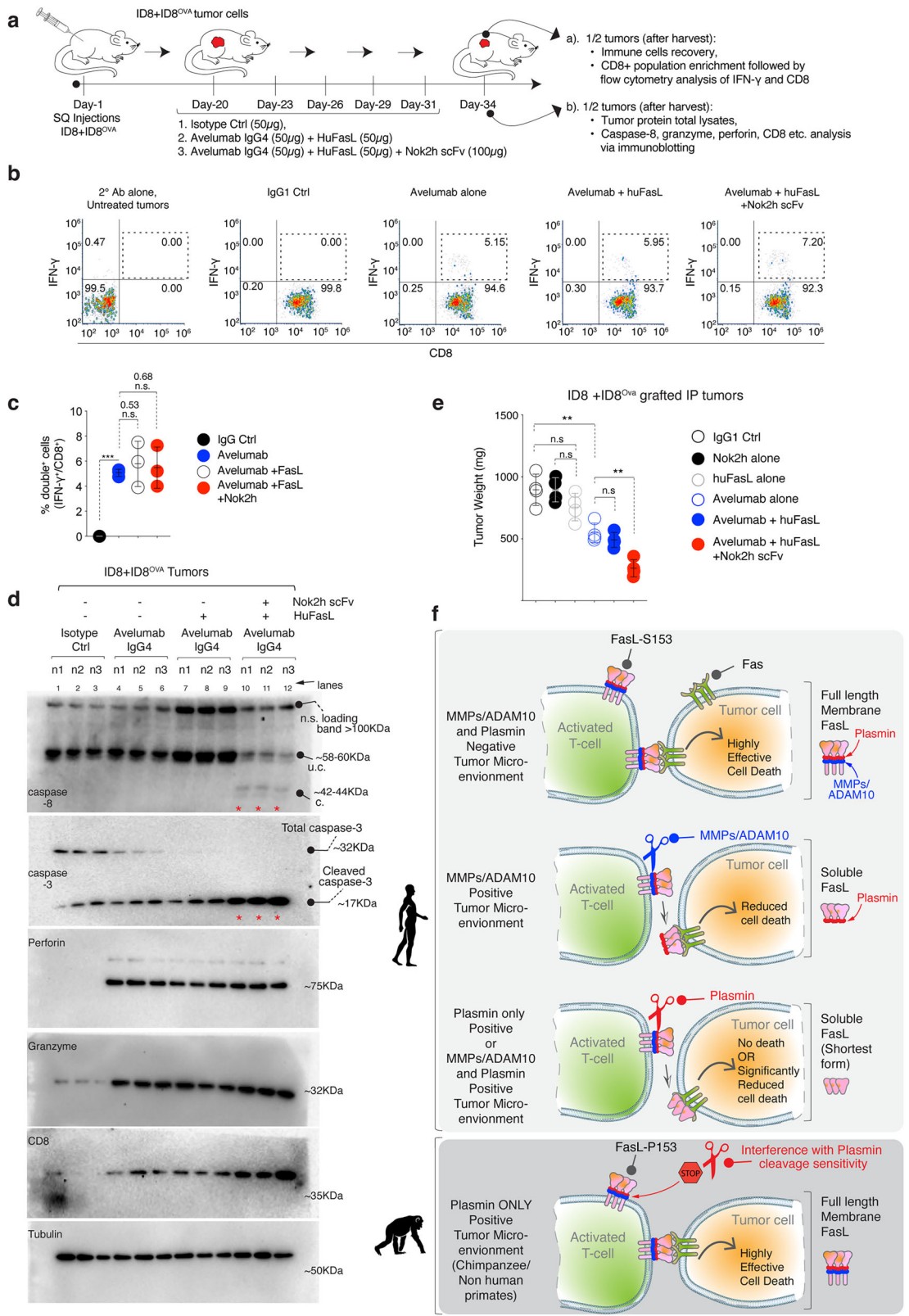

previously described[85]. 100 µg DNA was transfected using 1 µg/ml polyethyleneimine (PEI). After transfection, CHO cells were kept at 37 °C for 24 h, followed by shifting to 32 °C for 10–12 additional days. Cells were routinely fed (every 2nd day) with a 1:1 ratio of Tryptone feed and CHO Feed B (life technologies). After 10–12, supernatant from cultures was harvested and purified using either his-TRAP or protein-A affinity columns. Purified proteins were immediately buffer exchanged

in PBS using Amicon® Ultra Centrifugal Filter (Millipore) and stored at −80 °C until further use.

### Size exclusion chromatography
Various freshly purified FasL proteins [either 103–281 amino acids (HuFasL$^L$, RhFasL$^L$, HuFasL-S153P$^L$) or 134–281 amino acids (HuFasL$^S$, RhFasL$^S$, HuFasL-S153P$^S$) or 148–281 amino acids (HuFasL$^{ST}$, RhFasL$^{ST}$,

**Fig. 9 | Anti-PD-L1 and anti-Nok2h co-treatment rescues FasL signaling and improve overall antitumor function. a, b** Same as Fig. 8h-k, except animals were intraperitoneally (i.p.) injected either with 50 μg avelumab or avelumab +IgG1-huFasL or avelumab +IgG1-huFasL +Nok2h every 3rd day for a total of five doses. Harvested, sized matched tumors pooled by treatment group were distributed into two groups. From group-1, CD8$^+$ cells were enriched from the weight matched tumors (see "Methods"). Enriched CD8$^+$ T-cells from indicated groups were resti-mulated with anti-CD3 (OKT3) antibody for additional 2 h followed by intracellular IFN-γ expression analysis of CD8 gated cells using flow cytometry. **c** Percentage of CD8$^+$/ IFN-γ$^+$ double positive cells from three independent experiments. ($n = 3$, independent biological replicates from weight matched tumors, 3–6 animal used for tumor recovery for each treatment, Supplemental Fig. 13a for Gating strategy). IgG vs. Avelumab, ****$p$ = <0.0001; Avelumab Avelumab vs Avelumab + huFasL, nsp = 0.5344; Avelumab vs Avelumab + huFasL + Nok2h scFv, nsp = 0.6875; Unpaired, two-sided parametric t-test with no adjustments. **d** Same as (**a**, **b**), except

from the group-2, the harvested tumors total lysates were prepared and were analyzed for caspase-8, caspase-3, granzyme, and perforin. Representative data in blot is $n = 1$, however each treatment is represented with $n = 3$ animals. **e** Same as (**a–c**), except animals were intraperitoneally (i.p.) injected either with 50 μg of indicated antibodies every 3rd day until the tumor burden in control animals reached closed to the permissible IACUC size limit or if animal conditions were moribund. At the end of experiments average tumor weights was calculated and plotted from various treatment groups. ($n = 4$ mice) IgG vs. Nok2h scFv alone, nsp = 0.9988; IgG vs. huFasL alone, nsp = 0.1464; IgG vs. Avelumab alone, **$p$ = 0.0035; Avelumab vs Avelumab + huFasL, nsp= 0.2927; Avelumab vs Avelu-mab + huFasL + Nok2h scFv, **$p$ = 0.0016; Unpaired, two-sided parametric t-test with no adjustments. **f** Working model for human vs non-human primate FasL functional regulation by plasmin in tumor microenvironment. Error bars and data in (**c, e**) are presented as mean ± standard deviation (SD). In (**d**) u.c. indicates uncleaved, c. indicates cleaved.

HuFasL-S153P$^{ST}$)] post buffer exchange in PBS were immediately ana-lyzed using size exclusion chromatogram (Cytiva-AKTA Pure). An equal 10 μg concentration of each FasL was loaded onto a Superdex 200 10/300 GL gel filtration column. All samples were run at UV Absorbance at 280 nm and 1 ml/min flow rate as published[84]. Following the run completion, the elution profile was exported as an Excel file, and a chromatogram was developed.

### Plasmin in vitro cleavage assay
Plasmin was purchased from two different sources: Millipore (Catalog No. 527621-10U) and Innovative Research (IHUPLM1MG). Purified huFasL, RhFasL, and huFasL$^{S153P}$, as indicated in the text, were incu-bated either with 0.05–0.1 U range (Millipore) or 2 μg (Innovative research) in 100 mM sodium phosphate, 50 mM TRIS, 1 mM 6-aminohexanoic acid, pH 7.3 for indicated times at 37 °C for various times as indicated in the text. After incubation, 4× reducing SDS loading dye was directly added to the reaction mix, followed by boiling the samples at 95 °C for 5 min and loading them on SDS-PAGE followed by immunoblotting using anti-FasL or anti-Fc antibodies as indicated in the Figure legends.

### Adam10 cleavage assay
Recombinant HuFasL and RhFasL containing the extracellular FasL domain (aa 103–281) were incubated with recombinant human His-tagged ADAM10 (Catalog #: 936-AD, R&D Systems, USA) (1μ/20 μl reaction mixture) for indicated time at 37 °C in reaction buffer (25 mM Tris, 0.005% Brij-35; 2.5 μM ZnCl$_2$, pH 8.8) as published earlier[26]. After the indicated reaction time, proteins were mixed 4× reducing SDS loading dye was straight added to the reaction mix, followed by boiling of samples at 95 °C for 5 min and loading on SDS-PAGE and were analyzed by immunoblot with anti-His antibodies and anti-Flag anti-bodies (FasL). Since c-terminal His-tagged proteins were not func-tional, N-terminal His-tag proteins were synthesized, which were fully functional. However, after cleavage, the N-terminal target was way too small to capture even on 14% SDS Gel; hence, an anti-FasL antibody (cell signaling, 68405) was used for most blots. We also mapped the epitope of this antibody (data available), which was near the trimer-ization domain.

### Cell lines
All the human or murine tumor cell lines (colo-205, OVCAR-3, MDA-MB231, HeyA8, MDA-MB468, 4T1, MC38, ID8, ID8$^{OVA}$ etc.) were main-tained in DMEM and RPMI-1640 medium supplemented with 10% heat-inactivated fetal bovine serum (FBS), two mM glutamine, 100 U/ml penicillin, and 100 μg/ml streptomycin (complete medium) unless otherwise specified. Patient derived chemoresistant ovarian cancer cell lines (such as PDC1030, PDC1031 etc.) were kindly provided by Dr. Jeremy R. Chien, Department of Biochemistry and Molecular Medicine,

UC Davis. The HcMEC cells were kindly provided by Dr. Angie Gelli, Dept. of Pharmacology UC Davis. Primary Human Brain Microvascular Endothelial (HbMEC) Cells (ACBRI 376) were purchased from Cell Systems. Both of these cell lines were grown using EBM basal media and EGM-2 bullet kit (CC-4176, Lonza). Various cell lines were trypsi-nized and expanded as follows: After digestion, the cell suspension was neutralized with complete media and centrifuged for 5 min at 1500 rpm. The cell pellets were suspended in relevant DMEM/RPMI media and either expanded or seeded after counting using Countess II (Life technologies). Passaged cell lines were routinely tested for mycoplasma using MycoAlert Detection Kit (Lonza).

### Tumor control studies
The mice were housed under a 12-h light/dark cycle, at 68–78 °F with 30–70% humidity here at UC Davis Vivarium in Tupper Hall. 6–8 weeks old (Age), 20–25 g (Weight), both male and female (Sex) mice (NOD.Cg Prkdc$^{scid}$ Il2rg$^{tm1Wjl}$/SzJ also called as NSG), were used for tumor xenografts generation. All animal procedures were con-ducted in accordance with the University Animal Care and Use Committee (IACUC-UC Davis) and NIH approved protocols and conformed to the relevant regulatory standards. Weight and age (6–8 weeks old) matched animals were injected subcutaneously (SC) in their right flank with either colo-205, MDA-MB231, OVCAR3, or HeyA8 tumor cells (1 × e$^6$) in 100 μl Matrigel. When mice tumors were around ~100 mm$^3$, tumor-matched animals were randomly assigned and injected intraperitoneally (50 μg) either with rFOLR1, huFasL, RhFasL, or huFasLS153Pas indicated. Proteins were injected every third day for almost 30 days, after which tumors were harvested and weighed. In some studies, along with protein, either human plasmin-derived plasmin (52-762-110U, Sigma) or plasmin inhibitor (Pln-i) aprotinin (CAS 9087-70-1, Sigma) was used. Tumors during treat-ments were measured in two dimensions using a caliper, and after harvest, they were weighed as published[86] ($n = 4–6$ animals were used for each injection). The maximal total tumor burden permitted by UC DAVIS IACUC ethics committee is ≥10% of the animal's normal body weight. Measuring the mass of a tumor in vivo typically entails equating 1 cm$^3$ of tumor growth to 1 g of body weight. For example, a tumor measuring 2 × 2 × 2 cm (or 8 cm$^3$) is equivalent to a tumor mass of 8 g. If a mouse weighs 30 g, a tumor burden of 8 g is >10% of the animal's normal body weight and therefore meets a humane end-point. In all our experiments UC DAVIS IACUC ethics committee guidelines were followed, and maximal tumor size/burden was not exceeded beyond the permitted limits. In addition, following criteria for humane euthanasia of mice were followed as per ethics com-mittee guidelines: 1. Necrosis/ulceration/scabbing greater than 2 cm in any direction. 2. Chronic discharge present (greater than 2 days). 3. Active bleeding or deep tissue exposure. 4. Animal attending to lesion (recurrent scratching/biting of tumor). 5. Any sign of pain or

discomfort (hunched posture, painful response upon gentle tumor manipulation).

The $p$-values are determined by a two-tailed paired Wilcoxon Mann-Whitney test

## In vitro cell viability assays

Cell viability following various recombinant FasL treatments (as indicated in various figures), either alone or along with other treatments, was carried out using MTT cell proliferation assay kits as per manufactured protocols. Briefly, cells (indicated cells) were treated with increasing concentrations of various FasL (as indicated) along with relevant positive and negative controls for indicated times. For each cell-killing assay, the figures show the representative profiles from $n = 2$–3 with different cultured confluency.

## IC$_{50}$ determination

IC$_{50}$ values were calculated using MTT assays. The day before the assay, the cells were seeded in 96 well plates. The following the day, cultured cells were incubated for 48 h at 37 °C (5% CO$_2$) with increasing concentrations of the FasL WT or variants (as indicated in figure legends). Values obtained after reading the 96 well plates were normalized to the control group, and IC50 values were calculated using nonlinear dose-response regression curve fits using GraphPad Prism software. The final results shown in the histograms were obtained from three independent experiments. Whenever provided in the curves, the error bars show ±SEM.

## Western blotting

Cells were cultured overnight in tissue culture-treated six-well plates prior to treatment. After various FasL treatments (see legends) for the indicated time, cells were rinsed with PBS and then lysed with RIPA buffer supplemented with a protease inhibitor cocktail (Thermo Scientific). Spinning at 14,000 rpm for 30 min cleared Lysates and protein was quantified by Pierce BCA protein assay kit. Western blotting was performed using the Bio-Rad SDS-PAGE Gel system. Briefly, 20–30 μg of total protein was resolved on 10% Bis-Tris gels and then transferred onto the PVDF membrane. Membranes were blocked for 1 h at room temperature in TBS + 0.1% Tween (TBST) with 5% non-fat dry milk. Membranes were probed overnight at 4 °C with primary antibodies. Membranes were washed three times in TBST and then incubated with anti-rabbit or anti-mouse secondary antibodies (1/10,000 dilution, coupled to peroxidase) for 1 h at room temperature. Membranes were then washed three times with TBST, and Immunocomplexes were detected with SuperSignal West Pico Chemiluminescent Substrate (Thermo Fisher Scientific). Images were taken using a Bio-Rad Gel Doc Imaging system.

## Antibodies for immunoblotting

Most primary antibodies used were from cell signaling (or otherwise indicated as follows: Plasminogen Antibody #12657, PAI-1 (E3I5H) XP® Rabbit mAb #49536, uPAR (D7X2N) Rabbit mAb #12713, FasL (D1N5E) Rabbit mAb #68405, Fas (C18C12) Rabbit mAb #4233, His-Tag Antibody #2365, G4S Linker (E7O2V) Rabbit mAb (PE Conjugate) #38907, uPA (E2M6I) Rabbit mAb #15800, CD8 (D4W2Z) XP® Rabbit mAb #98941, Granzyme (E6Z6O) Rabbit mAb #98946, Perforin (E7D8R) Rabbit mAb #62550, Cleaved Caspase-8 (Asp374) (E6H8S) Rabbit mAb #98134, Caspase-3 Antibody #9662, His-Tag Antibody #2365, Myc tag antibody 9E10, Caveolin-1 Antibody #3238, N-Cadherin (D4R1H) XP® Rabbit mAb #13116, c-Met (D1C2) XP® Rabbit mAb #8198, PARP Antibody #9542.

## FasL neutralization

Whenever indicated throughout the manuscript text or in figure legends, the pre-neutralization of FasL (to interfere with plasmin cleavage) was carried out using anti-FasL antibodies such as Nok2-IgG2a, Nok2h-IgG2a, Nok2-scFv, Nok2h-scFv, 9F5-scFv, etc. as indicated in the text. For in vitro and in vivo studies, antibodies and indicated recombinant FasL proteins (human or murine or rhesus, etc.) were incubated together (either 1:1 or 1:2 ratio, as indicated) at 37 °C for 1 h shaking on a platform. As a control, indicated non-pre-neutralized antibodies were also incubated at 37 °C for 1 h shaking on a platform either with PBS alone or with recombinant non-specific proteins such as anti-FOLR1 scFv or random scFv. Following pre-neutralization, antibodies were either used in vitro for cell-killing assays, for cellular/tumor lysates generation (immunoblotting), or for in vivo tumor studies as indicated.

## Binding studies by SPR

Plasmin binding to humans and Rhesus FasL was carried out using Carterra's surface plasmon resonance (HT-SPR) technology, which utilizes high throughput and minimal sample requirements. Briefly, the human plasmin was covalently linked to an HC30M (30 nM linear polycarboxylate) chip surface (Carterra #4279) activated with a coupling running buffer (30 mM MES pH 5.5 + 130 mM NaCl and 0.05% Tween 20) followed by activation with 20 mM EDC + 5 mM S-NHS in 100 mM MES pH 5.5 for 7 min. The plasmin was diluted to 50 μg/mL in 10 mM acetate at indicated pH values and was immobilized on the chip for 20 min. The chip surface was then deactivated with a 1 M ethanolamine solution, pH 8.5, to inhibit further primary amine coupling. Next, the titrations of huFasL and RhFasL (his-tagged) were prepared starting at 1 μM (28 μg/mL) threefold dilutions serial. The kinetic series was run by first injecting eight buffer cycles, then the titration series of huFasL, then 6 more blank cycles, then the RhFasL titration 45-s baseline, 5-min FasL injection, and 15-min dissociation phase. The binding kinetics were carried out at 25 °C as per the manufacturer's recommendations. All data were fitted using the Kinetics software suite (Carterra) with a one-site model.

## Flow cytometry for membrane FasL cleavage

CHO-K cells were transfected with a pBOS-HFLD19 vector (kindly provided by Dr. Shigekazu Nagata) that lacks proline-rich 8–69 cytoplasmic amino acids but has been used by numerous studies[42,43] to study membrane FasL (FasL$^{Mem}$) cleavage by plasmin in co-cultures or cell killing in low or high c-MET expressing lines. The transient FasL$^{Mem}$ expressing CHO-K cells were co-cultured either with low c-Met expressing OVCAR3 or high c-Met expressing HbMEC endothelial cells or with PDC$^{1031}$ cells, as indicated in the figure legends. For flow cytometry studies, the mixed cell suspension (CHO-K-FasL$^{Mem}$ + PDC$^{1031}$ cells) was then incubated with anti-FasL primary antibodies for 1 h at 4 °C with gentle mixing. CHO-K non-transfected or PDC$^{1031}$ cells were negative control. For the mixing co-culture experiment, in control scenario CHO-K-FasL$^{Mem}$ + PDC$^{1031}$ cells were mixed on ice, while for plasmin cleavage scenario, they were incubated together for 4 h at 37 °C. Following wash with FACS buffer, the cells were then incubated with fluorescently labeled secondary antibody for 1 h. One experimental set was carried out using secondary antibody alone to get rid of the background. Cells were washed, and flow cytometry was performed using FACSCalibur. FCS Express (De Novo Software) and FlowJo analyzed the data.

## Flow cytometry of tumor isolated T-cells

The T-cells isolated from tumors (see TIL isolation method part) were analyzed for cell surface expression of PD-L1, FasL, Fas, CD3, CD8, CD4, IFN-γ, etc. was analyzed by flow cytometry. Briefly, overnight-grown tumors were trypsinized and suspended in FACS buffer (PBS containing 2% FBS). The single-cell suspension was then incubated with primary antibodies for 1 h at 4 °C with gentle mixing. Following wash with FACS buffer, the cells were then incubated with fluorescently labeled anti-mouse or anti-Rabbit secondary antibody (depending on the primary) for 1 h. Cells were washed, and flow cytometry was performed

using FACSCalibur. FCS Express (De Novo Software) and FlowJo analyzed the data.

## Plasmin activity assay

For the plasmin activity assay, a commercial kit that uses a colorimetric method was used to measure the activity (QuantiChrom Plasmin Assay Kit, DPLM-100). First, a standard was prepared using a series of plasmin dilutions by premixing it in a provided 6 μL of 100 mM pNA standard and 194 μL of assay buffer. Then, the provided known substrate was added as per the manufacturer's recommendations. The 96 well plate was tapped briefly by hand to mix the components. The mixture was later incubated at 37 °C for 60 min in aluminum foil to protect it from the light as per manufacturer recommendations, after which the plate was read at OD405nm on a plate reader (BioTek Synergy HTX Reader). Using the values of known plasmin concentration and corresponding normalized relative colorimetric values of the signals, the standard curve (filled black circle) was plotted. Next, in similar conditions, total membrane-enriched lysates from various indicated cell lines treated with ±uPA agonist antibody were incubated, and the relative colorimetric values from the cell lysates were plotted (red star = +uPA agonist A8 antibody, blue star = -uPA agonist A8 antibody) to determine the approximate plasmin activity in lysates in relation to the standard.

## Mechanical dissociation of tumors to obtain single-cell suspensions or lysates

Viable single cells from tumor tissues were isolated[87]. Briefly, after indicated FasL or antibody treatments (4–6 doses), mice were euthanized, and tumors were harvested using sterile scissors and forceps. After the excision of the tumor, they were minced into small pieces in sterile RPMI-1640 media using two single-edged razor blades. Small tumor pieces were passed through a 70 μm cell strainer in sterile RPMI-1640 media. A rubber plunger and syringe were used to mesh the dissociated cells through the cell strainer, and media containing dissociated cells was collected onto a sterile labeled conical tube. Dissociated tumor cells were subjected to flow cytometry (FACS) analysis as indicated for lysates generated for immunoblotting.

## Tumor infiltrated leukocytes (TILs) isolation by Ficoll-Paque density gradient centrifugation

TILs from indicated MC38 and ID8^OVA tumors were isolated as published here[88]. Briefly, after indicated antibody treatments, mice were euthanized. Next, tumors were harvested using sterile scissors and forceps. After excision, tumors were minced into small pieces in RPMI-1640 media using two single-edged razor blades. Small tumor pieces were transferred to a 70 μm cell strainer in RPMI-1640 media. A rubber plunger of a syringe was used to mesh the dissociated cells through the cell strainer, and cloudy media (that contained dissociated cells) was collected onto a sterile labeled 50 ml conical tube. Tubes were filled with 30 ml of RPMI-1640 media at room temperature (18–20 °C). Immediately before the addition of Ficoll−Paque media, single-cell suspension was well mixed with a 25 ml pipet. Thoroughly mixed 10 ml of Ficoll−Paque media was carefully poured into the bottom of the tube to form a layer of Ficoll−Paque below the cell suspensions without mixing the cell suspension. Tubes were centrifuged at 1025 g for 20 min at 20 °C with slow acceleration and without applying any brake. 20 ml of the upper layer of media was discarded to a waste bottle from the tube. A layer of mononuclear cells that contained leukocytes (CD3, CD4, or CD8 + T-cells) was transferred to a sterile labeled 50 ml conical tube using a sterile pipette, along with the remainder of the media above the Ficoll-Paque. The isolated TIL cells (in the mix of other mononuclear immune cells) were washed three times using 40 ml of complete RPMI media each time and were enriched for CD3 (or CD8) population using magnetic beads (MACS CD3, CD8 MicroBeads). After

the final wash and enrichment, isolated TILs were subjected to FACS analysis with fluorescently labeled CD3, CD8, FasL, IFN-γ, etc. antibodies as indicated in the manuscript text[89].

## Gating strategy for flow double positive flow cytometry data in Fig. 8 (CD3+FasL) and Fig. 9 (CD8 + IFN-γ).

The flow cytometry data analysis was carried out using FCS Express software (version 7). First, the intact cell population was identified by the forward (FSC) and side (SSC) scatter profiles, and the first gate was placed accordingly. Next, the singlet cell populations were determined and separated based on FSC-A vs FSC-H profile and subsequent the second gate was applied. Later among the singlets, live cells were identified on the basis of Zombie Aqua (Live/Dead marker) staining and taken for the final dual color density plot (Supplemental Fig. 13a).

## Nude and syngeneic tumor xenograft studies

All animal procedures were conducted under the accordance of University of California Davis IACUC approved protocols and conform to the relevant regulatory standards. For mice nude and syngeneic tumor studies, each mouse was counted as a biologically independent (sample). Regardless of gender, the mice were randomized before commencing the in vivo tumor studies. The investigators were not blinded to mice allocations to various treatment experimental procedures and during the outcome assessments. Details of mice strains, age and sex used are provided above. Briefly, 6–8 weeks-old (Age), 20–25-g (Weight), both male and female (Sex) mice were used for tumor xenografts generation, in vivo efficacy studies, imaging studies, TIL isolation studies as stated in the text. C56BL/6 (Jackson laboratories) mice stain was used for the immune sufficient studies using MC38 and ID8 cells. Immunodeficient NOD.Cg Prkdc^scid Il2rg^tm1Wjl/SzJ also called as NSG mice were used for immunodeficient tumor studies using indicated cell lines and patient-derived PDC[1031] cells. Various different tumor cell lines (as indicated in the figure legends) were used for tumor studies as in manuscript text. Weight and age (6–8 weeks old) matched mice were injected SC in their right flank with indicated cell lines in matrigel. Different cell number was injected as some cells were highly effective and some required higher density during xenografts. Tumor cells were mixed 100 μl volume with Matrigel. For antitumor efficacy studies, mice bearing ~100 mm³ tumors weight matched animals were randomly assigned into groups and injected (either 25 μg or indicated different dose) either intraperitoneally or intravenously (as indicated in Figure legends) three times per week with indicated FasL or antibodies in text and figure legends. Tumors were measured in two dimensions using a caliper method[86,90]. Tumor volume was calculated using the formula: $V = 0.5a \times b^2$, where a and b are the long and the short diameters of the tumor respectively. ($n = 4$–6 animals were used for each therapeutic antibody injection). The p values are determined by two-tailed paired Wilcoxon Mann-Whitney test. For syngeneic and surrogate tumor xenograft studies, most experiments were carried out using MC38 and ID8^OVA. MC38 and ID8^OVA are mouse epithelial colon and ovarian cancer line forms tumor in C57BL/6J. Similar to nude xenograft studies, 6–8-week-old female littermate of matched size and weight C57BL/6J mice were injected SC in their right flank with $0.5 \times 10^6$ MC38 or ID8^OVA cells lines in Matrigel. Both cells very highly consistently in forming tumors (19 out of 20) within ~2–3 weeks as published[91]. For tumor regression studies and TIL isolation studies mice bearing ~100 mm³ tumors were (after matching tumor size, number as indicated in legends) randomly assigned into groups and injected with a particular set of antibodies as indicated (50–100 μg dose) intraperitoneally two-three times per week (generally 5–6 doses or as indicated in figure legends). Most in vivo experiments were repeated three times or as indicated in figure legends. For anti-PD-L1 (avelumab-IgG4) Isotype IgG4 control were engineered and used. For Nok2h and 9F5 scFv, random scfv (or anti-FOLR1 scFv) with similar length G4S linker and similarly located his-tagged were engineered and

used. The *p* values are determined by two-tailed paired Wilcoxon Mann-Whitney test. For Biochemical analysis, signal cell isolation or TIL of tumors, mice were euthanized after indicated antibody treatment followed by tumor extraction[90,92].

## Third party rights

Figures 1a, 2a, c, f, 3a, b, d, 4a, c, d, e, 5h, k, 6a, e, g, h, l, 7j, 8f, h, 9a, f were drawn using adobe illustrator with institute/lab paid license. Supplementary Fig. 6c, 7a, c, d, 8d, 9c, 10b, and 12a were drawn using adobe illustrator with institute/lab paid license. Protein structures included in Figs. 1 b, c, 6i, k were analyzed using USCF chimera software, https://www.cgl.ucsf.edu/chimera/.

## Quantitation and statistical analysis

FlowJo V10 software was used to analyze data from flow cytometry analysis. All results were summarized and analyzed using a GraphPad Prism 9 software. Data, unless indicated otherwise, are presented as individual values or as mean ± SEM unless stated otherwise. In general, technical replicates were shown for in vitro experiments, a student *t* test was used for statistical analysis. It was also made sure that the same experiment was at least repeated once with a similar trend observed. When data from multiple experiments was merged into one figure, appropriate statistical analyses were performed using unpaired two-tailed *T*-tests with significance determined. For all the statistical experiments, *p* values, $p < 0.05$ (*), $p < 0.01$ (**), $p < 0.001$ (***) and $p < 0.0001$ (****) were considered statistically different. The *p* values are indicated by *, **, ***, **** symbols in the figures or figure legends wherever applicable. The n.s. (or nsp) in the figure legends indicate non-significant difference in *p* values.

## Statistics and reproducibility statement

Each experiment in the manuscript has been repeated independently. Most of the immunoblots has been repeated more than 2 or 3 times independently. All cell killing and animals' experiments have been either repeated biologically more than two times or were technically repeated >3 times or done from >3 independent cellular lysates/samples or have been done in more than >3 animals. For all the IgG1-FasL plasmin cleavage experiments the full-length blots are provided even in the main figures to show the detailed release of various Fc fragments.

## Reporting summary

Further information on research design is available in the Nature Portfolio Reporting Summary linked to this article.

## Data availability

The Source data are provided with this paper as a Source Data file. For additional information contact the corresponding author jtsingh@ucdavis.edu. Source data are provided with this paper.

## Code availability

No codes were generated in this manuscript.

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

## Acknowledgements

We are thankful for various UC Davis core facilities for assistance and members of Biological Use Authorization (BUA) committee and from the UC Davis Institutional Biosafety Committee (IBC) committee for assistance. All animal studies were approved by the Institutional Animal Care and Use Committee. We are highly thankful to Dr. Shigekazu Nagata (Osaka University, Japan) for providing FasL expression pBOS-HFLD19 vector. We are also thankful to Dr. Patrick Legembre and Dr. Laurent Devel for providing 9F5 VH and VL sequences. This work was supported by NCI/NIH grant (R01CA233752) and, U.S. DoD Ovarian Cancer Academy early career investigator award (OCRP: OC180412) to J.T.-S.

## Author contributions

Experimental methodology and data generation: B.E.N.W., T.M., F.F., D.B., M.S., O.P., S.B. and J.T.-S. Resources: Dr. Patrick Legembre and Dr. Laurent Devel for providing 9F5 VH and VL sequences, and Dr. Shigekazu Nagata (Osaka University, Japan) for providing FasL expression pBOS-HFLD19 vector. Help with editing and critical reading of the final manuscript: S.B., P.L., L.D., G.S.L. Study design: J.T.-S., Original draft: J.T.-S. Supervision and funding: J.T.-S. All Authors read and approved the final manuscript.

## Competing interests

The authors declare no competing interests.
