## [Transparent Peer Review file · Nature Communications]

Evolutionary Regulation of Human FasL by Plasmin in Solid Cancer Immunotherapy

Corresponding Author: Dr JOGENDER Tushir-SINGH

Version 0:

Reviewer comments:

Reviewer #1

(Remarks to the Author)

The authors describe a single amino acid substitution (P153S) that renders human FasL (and not in non-human primate FasL) susceptible to plasmin cleavage and a subsequent loss of its apoptotic function. The possible consequence of this is that targeting FasL as an anti-cancer therapy would be compromised if plasmin levels are high in the tumor. While this is an interesting observation, the manuscript is poorly written and very difficult to follow.

However, previous studies have already reported that FasL is cleaved by plasmin and this resulted in an increase in its proapoptotic function (ref 11 in manuscript; <https://pubmed.ncbi.nlm.nih.gov/18835034/>). In this manuscript, plasmin cleaved FasL loses its anti-apoptotic function. The authors do not discuss this opposite effect of plasmin cleavage.

The authors have not conducted experiments to generate plasmin within the tumor environment but rather have just added high levels of active plasmin. To strengthen the biological relevance the authors should undertake experiments to show that plasminogen activation by endogenous plasminogen activators (tPA or uPA) can initiate this process within the tumors themselves and lead to cleavage of FasL.

Do the authors know if the P153S mutation allows access of plasmin to a different cleavage site in FasL? The western blots in the manuscript (generally poorly described) are also unclear but might suggest this. In figure 1 panel f, treatment of huFasL with plasmin for 4 or 8 h causes almost complete loss of the 22kD FasL protein. Perhaps other cleavage sites are being acted upon. I suggest the authors mutate the RK cleavage site at position 144-145 to remove this plasmin cleavage site and then other sites can be revealed.

Another question I have relates to the cleavage of FasL by Adam-10. It is well known that Adam-10 cleavage of FasL also strongly influences the anti-apoptotic properties of FasL. So, when Adam-10 is present and promotes soluble FasL release, what additional effect will plasmin have?

In fig 2 the authors looked for changes in fibrinolytic parameters in tumor lines, specifically looking at PAI-1 and uPAR. It is unclear to me why uPAR was selected for this purpose (and not uPA). However, while some changes in the expression of these proteins is shown, there is no evidence that this is responsible for any change in plasmin activity. Why didn't the authors use plasmin activity assays to show plasmin actually is formed in under these conditions? Subsequent studies show that injection of aprotinin reduced tumor cell weight consistent with the authors hypothesis but direct evidence for plasminogen activation would be helpful. The authors also need to better describe Fig panel h (particularly HuFasL S153P/L153Q) which is not at all clear.

The authors also devote a lot of effort to related these findings to Textilin, a plasmin inhibitor from *Pseudonaja textilis*. While the parallels to FasL are apparent, I do not see how this relates to the modulation of FasL in vivo within tumors. The authors have not clearly integrated this aspect of the study and imply that textilin might modulate this process in vivo, but it is not clear to me it will ever be the case.

Minor:

Plasmin should be abbreviated as PIn, not Plg. Plg refers to plasminogen

Figure legends are poorly written. Are the western blots performed under reduced conditions?

Reviewer #2

(Remarks to the Author)

Having thoroughly examined the manuscript titled "Evolutionary Human FasL Variant Function is Regulated by Plasmin in Cancer," I find the premise of the study to be innovative and commend the authors for identifying a potential evolutionary mutation in human FasL that could have significant implications for cancer immunotherapy. The exploration of this mutation's impact on the susceptibility of FasL to plasmin cleavage is a novel approach to understanding interspecies differences in cancer treatment responses. However, I have several concerns that, if addressed, could significantly strengthen the manuscript.

- (1) The variability in tumor suppression across different tumor models suggests a complex relationship between FasL sensitivity and tumor type. While alternative explanations for these differences can be posited, the observed data indicate a need for a more robust correlation to establish the mutation's primary role in tumor response.
- (2) The use of human tumor cell lines implanted in severely immunodeficient NSG mice, while a common practice, presents limitations in accurately modeling the human immune response. The lack of a human immune system in these models may not fully capture the interactions between huFasL/RhFasL and human immune cells, stromal cells, and other somatic cells present in a clinical setting. It is recommended that the authors consider utilizing humanized mouse models with human immune systems to better replicate the clinical scenario.
- (3) The study's focus on the Pro153-Ser153 mutation's impact on FasL sensitivity to plasmin suggests a potential therapeutic avenue. However, the absence of validation experiments in a normal murine tumor model is noted. Introducing the Pro153-Ser153 mutation into murine tumor cells and assessing the efficacy of murine and mutated FasL, along with plasmin inhibitors, in immunocompetent mice could provide compelling evidence supporting the study's hypothesis.
- (4) The study primarily considers FasL expression in tumor cells, yet other non-tumor cells within the tumor microenvironment also express FasL. Clarification on whether the mechanisms of action are consistent across cell types is essential for a comprehensive understanding of the mutation's impact.
- (5) From an evolutionary perspective, the Pro153-Ser153 mutation is presumed to be advantageous. The manuscript would benefit from a discussion on why this mutation might be detrimental in the context of cancer therapy.
- (6) An exploration of whether human tumor tissues exhibit mutations at this site and their potential influence on tumor progression, prognosis, or response to other treatment regimens would be a valuable addition to the study.
- (7) The study does not delve deeply into the mechanisms underlying human FasL's increased sensitivity to plasmin, which is a critical aspect that warrants further investigation.
- (8) The presence of Ser153 in human FasL, an evolutionary distinction, raises questions about the safety of human FasL variants and plasmin inhibitors in tumor treatment. The study should address potential side effects and off-target effects on normal cells.
- (9) While the manuscript suggests that understanding genetic variations in FasL could improve CAR-T cell-based immunotherapies, it does not explore how this insight could be practically applied. Further research in this direction would be beneficial.

In conclusion, the manuscript presents an interesting and potentially impactful study on the role of an evolutionary substitution in human FasL and its regulation by plasmin in cancer. With the suggested revisions, I believe this manuscript could make a significant contribution to the field of cancer immunotherapy.

Reviewer #3

(Remarks to the Author)

The authors report the interesting finding that human FasL is cleaved by plasmin resulting in attenuation of its cytotoxic potential. They speculate that this could limit anti-tumoral T cell responses e.g. in CAR T cells therapies. The biochemical data concerning plasmin-mediated processing of human versus rhesus FasL, which is less efficiently processed, are convincing and straightforward. It should be clear stated that that FasL cleavage by plasmin has already been reported by Bajou et al. cited in the manuscript.

The discussion of the functional data is however not fully convincing:

Membrane FasL has much higher cytotoxic activity compared to proteolytically processed soluble FasL. Thus, an obvious interpretation would be that plasmin protects tumors from cytotoxic Fas activity by reducing the amount of the highly cytotoxic membrane FasL molecules. In this respect, I miss analysis of the effect of plasmin etc on the expression of human and rhesus membrane FasL. Ditto, I feel it is mandatory to evaluate the effect of plasmin on the cytotoxic action and processing of membrane FasL.

The self-assembly of FasL is not only driven by the stalk region, dedicated by the authors as trimerization region. The TNF

homology domain which is still functional present in plasmin-processed FasL is even more relevant for this as it is obvious from figure 2c.

It is known that oligomerization of soluble FasL trimers results in strong enhancement of their cytotoxic activity. Therefore, although soluble stalk-less and stalk-containing FasL variants may differ in their limited cytotoxic activity they still not reach the specific activity of oligomerized soluble FasL trimers which might better mimic the activity of memFasL.

In view of the considerations above, the animal experiment shown in figure 2g and 2h possibly targets the secondary question of the different residual cytotoxic activity of different soluble FasL molecules instead the, in my opinion, more obvious question whether plasmin-dependent reduction of membrane FasL of T cells hinders the latter from tumor killing. Indeed, in accordance with the notion that all the soluble FasL are poorly active, when compared with relevant benchmarks, such as memFasL or oligomerized soluble FasL, the authors injected mice with 50 ug of their soluble FasL molecules every three days while other reported that already 5 ug of oligomerized soluble FasL kills mice within 3 hours (e.g. Jost et al., 2009). – High liver toxicity is considered as the hallmark of systemic CD95 activation!

It could also be discussed that soluble FasL may efficiently promote cell-death independent activities not requiring oligomerization by e.g. calcium signaling (Devel et al., 2022).

I miss discussion of the existing literature concerning the activity of different forms of FasL including soluble FasL variants varying with respect to their “stalk content” and a stalk-related tendency to aggregate (without completeness e.g. Suda et al.; Berg et al.; Schneider et al.; Oyaizu et al.; Schulte et al.; Herrero et al.; O'Reilly et al).

Jost PJ, Grabow S, Gray D, McKenzie MD, Nachbur U, Huang DC, Bouillet P, Thomas HE, Borner C, Silke J, Strasser A, Kaufmann T. XIAP discriminates between type I and type II FAS-induced apoptosis. *Nature*. 2009 Aug 20;460(7258):1035-9. doi: 10.1038/nature08229.

Devel L, Guedeney N, Bregant S, Chowdhury A, Jean M, Legembre P. Role of metalloproteases in the CD95 signaling pathways. *Front Immunol*. 2022 Dec 5;13:1074099. doi: 10.3389/fimmu.2022.1074099.

Suda T, Tanaka M, Miwa K, Nagata S. Apoptosis of mouse naive T cells induced by recombinant soluble Fas ligand and activation-induced resistance to Fas ligand. *J Immunol*. 1996 Nov 1;157(9):3918-24. PMID: 8892623.

Berg D, Lehne M, Müller N, Siegmund D, Münkkel S, Sebald W, Pfizenmaier K, Wajant H. Enforced covalent trimerization increases the activity of the TNF ligand family members TRAIL and CD95L. *Cell Death Differ*. 2007 Dec;14(12):2021-34.

Schneider P, Holler N, Bodmer JL, Hahne M, Frei K, Fontana A, Tschopp J. Conversion of membrane-bound Fas(CD95) ligand to its soluble form is associated with downregulation of its proapoptotic activity and loss of liver toxicity. *J Exp Med*. 1998 Apr 20;187(8):1205-13.

Oyaizu N, Kayagaki N, Yagita H, Pahwa S, Ikawa Y. Requirement of cell-cell contact in the induction of Jurkat T cell apoptosis: the membrane-anchored but not soluble form of FasL can trigger anti-CD3-induced apoptosis in Jurkat T cells. *Biochem Biophys Res Commun*. 1997 Sep 18;238(2):670-5.

Schulte M, Reiss K, Lettau M, Marezky T, Ludwig A, Hartmann D, de Strooper B, Janssen O, Saftig P. ADAM10 regulates FasL cell surface expression and modulates FasL-induced cytotoxicity and activation-induced cell death. *Cell Death Differ*. 2007 May;14(5):1040-9.

Herrero R, Kajikawa O, Matute-Bello G, Wang Y, Hagimoto N, Mongovin S, Wong V, Park DR, Brot N, Heinecke JW, Rosen H, Goodman RB, Fu X, Martin TR. The biological activity of FasL in human and mouse lungs is determined by the structure of its stalk region. *J Clin Invest*. 2011 Mar;121(3):1174-90.

O' Reilly LA, Tai L, Lee L, Kruse EA, Grabow S, Fairlie WD, Haynes NM, Tarlinton DM, Zhang JG, Belz GT, Smyth MJ, Bouillet P, Robb L, Strasser A. Membrane-bound Fas ligand only is essential for Fas-induced apoptosis. *Nature*. 2009 Oct 1;461(7264):659-63.

Version 1:

Reviewer comments:

Reviewer #1

(Remarks to the Author)

The authors have provided a detailed response to my earlier comments. The manuscript is significantly longer as a result. I

am for the most part satisfied with the authors responses to my questions and that FasL contain a unique plasmin cleavage site that is present in human but not non-human FasL. A couple of questions remain:

Fig 3d: the authors claim that the 144AA145 mutation completely abolished the 31kD fragment release. However there is a band migrating slightly faster than 31kD (~25kD) that the authors state to be "non-specific". How can the authors be sure of this and that this is not a band produced from another cleavage site?

Western blots (in general). The description of the western blots is insufficient. At various times, the authors conducted western blots using "non-reducing and partly denatured gels". What does this mean? Samples are either denatured or not.

Reviewer #2

(Remarks to the Author)

I'm glad to receive the authors' point-by-point response. After looking at the new data they provided, I've found several areas that could be improved.

1. Issues with CD3 Staining and Gating in Fig8 i/j: There seem to be some problems with the CD3 staining and gating in Fig8 i/j. In the FACS plots, the cells shown should include both CD3-positive and -negative populations. In the secondary group (I assume it's the control group stained with CD3 and FasL antibodies), all cells are CD3-negative. In the IgG1 Ctrl group, there are both CD3-positive and -negative populations. However, in the Avelumab IgG4 group and Avelumab IgG4 + Nok2h scFv group, almost all cells are CD3-positive. I'm curious about the reasons for these differences among the last three groups. Why are the last two groups mainly composed of CD3-positive T cells, while the IgG1 Ctrl group has two distinct populations? And in the IgG1 Ctrl group, why isn't the separation between CD3-positive and -negative cells clear? If all groups are gated on the CD3+ T-cell population, how does the CD3-negative population in the IgG1 Ctrl group exist?

2. Problems with CD3 Staining and Gating in Fig9b: Similar issues exist in the CD3 staining and gating of Fig9b. There are clearly two populations of CD3-stained cells. But with the current +-shaped gate, the CD3-positive population is further divided into two parts (the left part is classified into the negative group), making the CD3-negative cells appear as two distinct populations. Although the authors set the positive gate based on the Secondary group, this gating method doesn't work well in this case. Similarly, the IFN gate also splits an obvious cell population into two, which is not ideal. I suggest the authors use an irregular gate to solve this problem.

3. Scarcity of CD3-Negative Populations in Fig8 i/j and Fig9b: In Fig8 i/j and Fig9b, the CD3-negative populations are surprisingly small. I'm wondering where the other immune cells have gone. It would be great if the authors could explain this.

4. Difference in FasL-Negative Populations in Fig8 i/j: In Fig8 i/j, the FasL-negative population in the IgG1 Ctrl group looks quite different from the other two groups. Could the authors provide an explanation for this difference? It might help us better understand the data.

5. Suggestion for Fig9b: For the FACS analysis in Fig9b, I suggest the authors create a statistical graph similar to Fig8k. This would make the data more intuitive and easier to compare.

6. Typo in FACS Staining Controls: In Fig8i/j and Fig9b, the FACS staining control is labeled as "secondary" in some places, but it should be "secondary". Please correct this typo to avoid confusion.

7. Inconsistency in Antibody Description: The authors' team treated the mice with the anti-PD-L1 (avelumab-IgG4) antibody. However, in the main text, it's written as "PD-L1 treated". Usually, this kind of expression implies treatment with the PD-L1 protein, not the antibody. It would be better to use a more accurate description to avoid misunderstandings.

Reviewer #3

(Remarks to the Author)

The authors have added a significant amount of data and convincingly addressed my concerns about the initial submission. The discussion has been significantly revised and is now balanced.

Version 2:

Reviewer comments:

Reviewer #1

(Remarks to the Author)

The authors have satisfactorily responded to my earlier comments.

Reviewer #2

(Remarks to the Author)

The authors have addressed all my previous concerns in the revised manuscript.

Point-by-point response to Reviewers comments

Reviewer #1

Point 1. Previous studies have already reported that FasL is cleaved by plasmin and this resulted in an increase in its proapoptotic function(ref 11 in manuscript; <https://pubmed.ncbi.nlm.nih.gov/18835034/>). In this manuscript, plasmin cleaved FasL loses its anti-apoptotic function. The authors do not discuss this opposite effect of plasmin cleavage.

Response: We appreciate the Reviewer's comment and fully agree that gain in apoptosis has been described to plasmin cleaved FasL in endothelial cells. Indeed, the study published in cancer cells ([https://pubmed.ncbi.nlm.nih.gov/18835034:PMID: 18835034](https://pubmed.ncbi.nlm.nih.gov/18835034:PMID:18835034)), was very insightful for our whole differential huFasL regulation discovery. We are aware of the opposite effect in endothelial cells vs tumor cells. Kindly see Figure 6, Supplemental Fig 10, and the discussion section; as we have compared ovarian tumor cells and endothelial cells (hBMEC cells used in PMID: 18835034) side by side and provided much additional data to differentiate the outcome in tumor cells vs. endothelial cells

We also want to point out that endothelial cells are very different than epithelial cells in terms of architecture and function. Indeed, in >80% of solid tumors, the epithelial cell undergoes transformation. Location-wise, endothelial cells are at the lining of blood vessels and lymphatic vessels, while epithelial cells line outer surfaces. Also, not only c-Met but various surface receptors, ECM proteins, and tight junctions are differently expressed and regulated in endothelial and epithelial cancer cells.

Notably, cMet (HGFR) is highly expressed on the surface of endothelial cells (Fig 6m, n) and is essential in endothelial cell proliferation and tube formation (doi: 10.1172/JCI84876). Interestingly, cMet has an opposing role in endothelial cells vs epithelial tumor cells. cMet has been shown to engage FasL (PMID: 17704785 and please data in Fig 6 i-l with YLGA mutant). Further, independent of FasL engagement, previous studies have established that cMet-Fas interactions limit canonical death-inducing signaling complex (DISC) clustering and caspase-8 in response to FasL and Fas agonist antibodies in endothelial cells (DOI: 10.1161/01.HYP.0000167991.82153.16). On the contrary, cMET-driven inhibition of Fas oligomerization in tumor cells is poorly documented.

To investigate the noticeable differences, we treated tumor cells (OVCAR-3) and endothelial cells (HbMEC) either with his-tagged soluble FasL (adam10 or plasmin cleaved: see FasL^S and FasLST schematic in Fig 6a) or by mixing CHO-K cells transiently expressing membrane FasL^{Mem} (Supplemental Fig 10) with OVCAR-3 or HbMEC cells. 2hrs later, we analyzed Fas clustering by running the total lysates in non-reducing and partly denaturing gels, followed by Fas immunoblotting as published by us for DR5 and Fas (DOI: 10.15252/emmm.202012716, DOI: 10.1038/s41418-023-01229-7). We also analyzed the FADD protein, the key protein to confirm DISC clustering. We observed opposite Fas and FADD clustering in OVCAR-3 and HbMEC cells (Supplemental Fig 10d-e). In OVCAR-3 cells, both membrane (FasL^{Mem}) and ADAM10 cleaved Fas (FasL^S) clustered Fas and FADD, while in HbMEC cells, both soluble FasL (ADAM10: FasL^S or plasmin: FasLST) partly clustered Fas and FADD. FasLST (plasmin cleaved) showed a different oligomerization pattern, as evident in endothelial cells' slightly shorter Fas and FADD clusters (Supplemental Fig 10d-e). We observed differential caspase-8 activation and caspase-3 cleavage (Supplemental Fig 10f). Strikingly, in agreement with cMet mediated negative regulation of Fas, treatment of HbMEC cells with c-Met blocking antibody, MetMab (a monovalent anti-c-Met: DOI: 10.1158/0008-5472.CAN-07-5960) selectively enhanced HbMEC killing (but Not of OVCAR-3) even higher than FasLST treated cells killing profile (Supplemental Fig 10c). These results explain the differential regulation of Fas signaling in endothelial cells. Although the cancer cell manuscript specifically focused on plasmin regulation (<https://pubmed.ncbi.nlm.nih.gov/18835034/>), our data indicate that even MMP cleaved soluble FasL will instigate cell death of endothelial cells. Hence, our data supports the differential regulation in endothelial vs tumor (or jurkat or lymphocytes). All these points are incorporated into the discussion section.

Point 2. The authors have not conducted experiments to generate plasmin within the tumor environment but rather have just added high levels of active plasmin plasmin. To strengthen the biological relevance the authors should undertake experiments to show that plasminogen activation by endogenous plasminogen activators (tPA or uPA) can initiate this process within the tumors themselves and lead to cleavage of FasL.

Response: We appreciate the reviewer's comment and agree that most experiments (except a couple) were carried out by adding exogenous plasmin in the original submission. Kindly see the revised manuscript, which has significant new data to support our hypothesis and findings.

In the revised manuscript, we have conducted experiments to test plasmin activity in terms of FasL cleavage by the plasminogen system in cell supernatants (Fig. 7j-k, Fig. 7h, Supplemental Fig 12 a, b). In addition, we have used a commercial plasmin activity assay kit and the previously described endogenous plasminogen activator uPA agonist antibody A8 (Fig 7g, Supplemental Fig 10c-d) using a few tumor cell lines. When tested in vitro assays and on cells against colo-205 and PDX cells, although both expressed uPA and plasmin pathway components, the addition of A8 scFc enriched the uPA-mediated plasmin function, as evident with the plasmin activity assay and tumor data. In addition, tumor data in Fig 8 and Fig 9 with immune-competent mice models harboring ID8 ovarian tumors support T-cell expressed membrane FasL cleavage by plasmin in plasminogen-system expressing ID8^{OVA} tumors.

Point 3. Do the authors know if the P153S mutation allows access of plasmin to a difference cleavage site in FasL? The western blots in the manuscript (generally poorly described) are also unclear but might suggest this. In figure 1 panel f, treatment of huFasL with plasmin for 4 or 8 h causes almost complete loss of the 22kD FasL protein. Perhaps other cleavage sites are being acted upon. I suggest the authors mutate the RK cleavage site at position 144-145 to remove this plasmin cleavage site and then other sites can be revealed.

Response: We strongly agree with the reviewer's insightful comment that the P153S mutation could allow Plasmin to access a different cleavage site in FasL.

In response, we provided SPR binding data of Plasmin with human FasL (P153S) and Rhesus FasL (S153P), supporting that bindings remained unchanged. Kindly see Fig 2e, Supplemental Fig 4, and 5 in the revised manuscript).

We also fully agree with the reviewer that the complete loss of the 22kD FasL signal in the western blots might suggest multiple cleavage sites and random cleavage.

To address this issue that ¹⁴⁴RK¹⁴⁵ residue is the only plasmin cleavage site in close vicinity of evolutionary S153 mutation (in the loop of ¹⁴⁴RK¹⁴⁵ connecting beta-strand), we generated Random IgG1-Fc conjugated FasL (IgG-FasL, kindly see data in Fig 3 and Fig 4). IgG1 hinge region was engineered with an optimal plasmin cleavage site (Fig 3), and the CH3 end of IgG1 was genetically ligated with either human FasL, Chimpanzee FasL, or Monkey FasL using a G4S linker.

Reengineered IgG-FasL now contains two plasmin cleavage sites: one in IgG hinge and the other in FasL (Fig 3a, sequences are shown, and scissors represent plasmin cleavage sites). Compared to IgG heavy chain, which is 50KDa, IgG-FasL heavy chain would be ~71KDa (See box "A", in Fig 3b). If plasmin cleavage happened only at the IgG1 hinge (see schematic Scissor 1), a ~47KDa fragment with CH2-CH3-linker-FasL will be released (See box "B" in Fig 3b). If plasmin cleavage happened only at the FasL site (see schematic Scissor 2), another ~47KDa fragment with VH-CH1-CH2-CH3 and a small linker will be released (See box "C" in Fig 3b). If plasmin cleavage happened at both places (see a schematic of Scissor 1 and Scissor 2), a ~31KDa fragment with CH2-CH3-small linker containing fragment will be released (See box "D" in Fig 3b). The latter (~31KDa fragment) will decisively indicate the cleavage of FasL at R144-K145. Importantly, all released fragments, plus Intact 75KDa heavy chain, would contain intact Fc (CH2-CH3) and will be easily detected on SDS-immunoblot by anti-human Fc antibody. As expected, only IgG-FasL with huFasL (linked to CH3 domain) showed significant release of ~31KDa fragment, further reaffirming selective cleavage of huFasL by Plasmin (Fig 3c). If cleavage had happened at other sites in FasL, other fragments larger than ~31KDa and smaller than ~47KDa should have been visible on the gel. However, that is not the case. These results conclusively confirm only one cleavage site, like at R144-K145. A non-specific band is evident in all lanes (including untreated), shorter than ~31KDa.

We also fully agree with the reviewer's suggestion to mutilate the RK cleavage site at positions ¹⁴⁴RK¹⁴⁵. In response, we generated another IgG-huFasL with the ¹⁴⁴RK¹⁴⁵ being mutated to ¹⁴⁴AA¹⁴⁵ and tested it in the same

assay (Fig 3d). As evident in Fig 3d, the ¹⁴⁴AA¹⁴⁵ mutation totally abolished ~31KDa fragment release. These results strongly support the ¹⁴⁴RK¹⁴⁵ residue in close vicinity of the evolutionary S153 mutation as a plasmin substrate. An overlay of an immunoblot with a membrane and a size ladder (left) marker are also shown in the Supplemental figures 6b, 6c, 7b etc.

Point 4. Another question I have relates to the cleavage of FasL by Adam-10. It is well known that Adam-10 cleavage of FasL also strongly influences the anti-apoptotic properties of FasL. So, when Adam-10 is present and promotes soluble FasL release, what additional effect will plasmin have?

Response: We appreciate reviewers' comments. Yes, we are aware and agree with the Adam-10 mediated cleavage of FasL influencing the anti-apoptotic properties of Fas. We have also included a reference to the latter in the manuscript. Our findings indicate that in the differential tumor microenvironment conditions where ADAM10 is not expressed (such as in the tumor data of ovarian ID8^{OVA} tumors in (Fig 8 and 9 and Supplemental figures), plasmin is expressed, and the latter regulates FasL. As shown in Supplemental Figure 12d, a pan ADAM10/17 did not reverse caspase-8 activation in ovarian ID8^{OVA} tumors, but plasmin cleavage inhibiting antibodies such as Nok2, Nok2h, and 9F5 scfv (provided by a FasL lab and collaborator) activated FasL-dependent caspase-8. Hence, although both reduce FasL activity, context matters in a particular tumor microenvironment. We would also like to point out that the literature is little muddy in terms of soluble FasL being completely ineffective or not (kindly see the discussion section). For example, there are studies that indicate that adam10 or MMP cleaved soluble FasL could still have cell death activity (although reduced). Please see the discussion section. At the same time, there are others that suggest that adam10 or MMP-cleaved soluble FasL completely loses activity. For example, studies such as PMID: 7536672 and PMCID: PMC3049393 suggest that soluble MMP-10 or adam10 cleaved partial stalk containing FasL is cytotoxic, while studies such as PMID: 9299572 indicate a total loss in activity. Furthermore, the study in the cancer cell (PMID: 18835034) indicates opposite of what is shown for membrane FasL (PMID: 9299572). We believe these observed differences in the cell death by different published studies are due to differential regulation of FasL by membrane proteins (such as c-met, DR5, etc. or others published) interfering with FasL aggregation and its FasL clustering function. Regardless, all these references are included in the manuscript and discussion section is included to state the latter.

Point 5: In fig 2 the authors looked for changes in fibrinolytic parameters in tumor lines, specifically looking at PAI-1 and uPAR. It is unclear to me why uPAR was selected for this purpose (and not uPA). However, while some changes in the expression of these proteins is shown, there is no evidence that this is responsible for any change in plasmin activity. Why didn't the authors use plasmin activity assays to show plasmin actually is formed in under these conditions? Subsequent studies show that injection of aprotinin reduced tumor cell weight consistent with the authors hypothesis but direct evidence for plasminogen activation would be helpful. The authors also need to better describe Fig panel h (particularly HuFasL S153P/L153Q) which is not at all clear.

Response: We appreciate the reviewer's comment and totally agree with missing the blots of not testing uPA instead of uPA.

Kindly see the data in Fig 8E, western blot. We have also used a uPA agonist antibody (see Supplemental Fig 11c, d) and an endogenous plasmin activity assay (Fig 7g, h, i, j, k). In addition, a significant amount of new data is provided on using plasmin cleavage-blocking antibodies in vitro (see Fig 5) and in vivo data (Fig 8, 9).

The HuFasL S153P/L153Q is typo mistake. It should be S153P/L143Q which has been updated in the revised manuscript. The point with the S153P/L143Q double mutant is that mutation of L-Q was insignificant at 143 positions but S153P is significant in efficacy.

Point 6. The authors also devote a lot of effort to relate these findings to Textilin, a plasmin inhibitor from *Pseudonaja textilis*. While the parallels to FasL are apparent, I do not see how this relates to the modulation of FasL in vivo within tumors. The authors have not clearly integrated this aspect of the study and imply that textilin might modulate this process in vivo, but it is not clear to me it will ever be the case.

Response: We appreciate reviewer's comment that Textilin and FasL parallels are apparent but unnecessary for the manuscript considering that we have made use of plasmin cleavage-blocking antibodies (see Fig 5-9). Hence, upon reviewers suggestion we have removed the Textilin part from the manuscript.

Minor:

Plasmin should be abbreviated as Pln, not Plg. Plg refers to plasminogen

Response: We appreciate reviewers' suggestion. Wherever plasmin abbreviation was used we have updated it Pln instead of Plg

Figure legends are poorly written. Are the western blots performed under reduced conditions?

Response: We appreciate reviewers' comment. If western blots were performed in reducing and non-reducing conditions, we have stated it in figure legends.

Reviewer #2

Having thoroughly examined the manuscript titled "Evolutionary Human FasL Variant Function is Regulated by Plasmin in Cancer," I find the premise of the study to be innovative and commend the authors for identifying a potential evolutionary mutation in human FasL that could have significant implications for cancer immunotherapy. The exploration of this mutation's impact on the susceptibility of FasL to plasmin cleavage is a novel approach to understanding interspecies differences in cancer treatment responses. However, I have several concerns that, if addressed, could significantly strengthen the manuscript.

(1) The variability in tumor suppression across different tumor models suggests a complex relationship between FasL sensitivity and tumor type. While alternative explanations for these differences can be posited, the observed data indicate a need for a more robust correlation to establish the mutation's primary role in tumor response.

Response: We appreciate reviewers' comments supporting our data that there is variability in tumor suppression in tested models due to P153S mutation. Kindly see the revised manuscript with a significant amount of new in vitro and in vivo data, including data from immune-competent models, which firmly established a more robust correlation to strongly support the P153S mutation's primary role in plasmin cleavage and, ultimately, tumor response.

(2) The use of human tumor cell lines implanted in severely immunodeficient NSG mice, while a common practice, presents limitations in accurately modeling the human immune response. The lack of a human immune system in these models may not fully capture the interactions between huFasL/RhFasL and human immune cells, stromal cells, and other somatic cells present in a clinical setting. It is recommended that the authors consider utilizing humanized mouse models with human immune systems to better replicate the clinical scenario.

Response: We appreciate the reviewers' comments and fully agree that severely immunodeficient NSG mice present limitations in accurately modeling and supporting the data. We also understand the reviewer's recommendations of utilizing mouse models with immune systems, which we have done in the revised manuscript. Kindly see Figures 8 and 9 and supporting Supplemental figures. We also agree with the reviewers' recommendations for using humanized mice models. However, the latter is beyond the scope of the current study, particularly for the reasons/their limitations as follows:

Based on the literature, no single humanized mouse model fully recapitulates various aspects of the tumor-immune interaction landscape within the TME. One of the most studied models for immuno-oncology research is the Hu-SRC humanized mice model. The latter is centered around injecting CD34+ hematopoietic stem cells (HSC). This model often does not fully recapitulate a human T-cell population's true diversity and functionality. Different research groups have shown issues in terms of generating differential outcomes when testing immune checkpoint inhibitors, depending on the existence of donor CD34+ HSPCs variability testing. Various groups have seen inconsistent efficacy responses to immunotherapies, primarily due to differential immune cell reconstitution rates. In other studies, patient cell-derived xenografts grew well in the engrafted Hu-SRC mice and also detected human T cell infiltration in all tumors; however, the numbers of infiltrated cells varied substantially between different tumors (PMCID: PMC7174072, PMID: 23931270, PMCID: PMC7783739, PMID: 36635480, etc.). Most researchers are focused on optimizing humanized mice models for a particular set of optimal therapeutic testing, and since there are no humanized mice described that fully capture FasL-mediated bystander or direct killing by T-cells, optimizing a humanized mouse to test the latter is beyond the scope of the

current study. Considering that our novel evolutionary findings are not focused on testing the efficacy of a therapeutic antibody but rather are centered around the FasL mechanistic regulation by plasmin in the tumor-immune contest, we did make use of well-established murine-syngeneic ovarian immune-competent mice models as per reviewers' suggestions. Kindly see the data in Fig 8 and 9. The provided data is robust and support the scientific premise of the study.

(3) The study's focus on the Pro153-Ser153 mutation's impact on FasL sensitivity to plasmin suggests a potential therapeutic avenue. However, the absence of validation experiments in anormal murine tumor model is noted. Introducing the Pro153-Ser153 mutation into murine tumorcells and assessing the efficacy of murine and mutated FasL, along with plasmin inhibitors, inimmunocompetent mice could provide compelling evidence supporting the study's hypothesis.

Response: We highly appreciate the reviewers' insightful comments, which were very helpful in reevaluating the mechanism in immunocompetent animal tumor settings. A significant number of new animal tumor data sets using syngeneic MC38 (negative for harboring plasminogen system components) and ID8 ovarian tumors (positive for harboring plasminogen system) are provided. Kindly see the data in Fig 8 and Fig 9 and the Supplemental figures, where we have tested the FasL cleavage by plasmin in tumor-infiltrated T-cells. Significantly, the latter was inhibited with plasmin cleavage interfering antibodies, as described by strong cleavage inhibition data sets in Fig 5. We have also carried out comprehensive biochemical tumor analysis in terms of selective caspase-8 activation only in the presence of plasmin cleavage interfering antibodies using immunocompetent animal tumor settings to provide compelling evidence supporting the study's hypothesis of the evolutionary significance of Pro153-Ser153 mutation in FasL.

(4) The study primarily considers FasL expression in tumor cells, yet other non-tumor cells within the tumor microenvironment also express FasL. Clarification on whether the mechanisms of action are consistent across cell types is essential for a comprehensive understanding of the mutation's impact.

Response: We appreciate reviewers' comments. However, we would like to clarify to the reviewer that our study DOES NOT primarily consider FasL expression in tumor cells. Indeed, we indicate that FasL expression on immune cells is under the onslaught of plasmin (and other MMPs) in the tumor microenvironment. We provide data using FasL expressed on murine T-cells (Fig 8-9) and exogenous addition of FasL to support the consistency in the T-cell manner. Indeed, using anti-PD-L1 antibodies, our data support the limited efficacy of immunotherapies in solid tumors (ID8-ovarian) expressing plasminogen system. We also used membrane FasL expressing CHO-K in coculture with plasmin-expressing tumor cells to support the consistency of the phenomenon in other non-tumor cells. Additional data has been added using endothelial cells (Fig 6, 7, Supplemental Fig 10). A whole new discussion section has been added to hypothesize the significance of the potential of FasL evolutionary mutations in contributing to larger human brain size vs primates.

(5) From an evolutionary perspective, the Pro153-Ser153 mutation is presumed to be advantageous. The manuscript would benefit from a discussion on why this mutation might be detrimental in the context of cancer therapy.

Response: We highly welcome reviewers' comments and insightful suggestions. In response, a whole new paragraph in the discussion focuses on brain size differences and potentially evolutionary perspectives. Based on the literature (all the references provided in the discussion), the Pro153-Ser153 mutation in FasL is potentially advantageous in increasing human brain size (as the expense of higher susceptibility to cancer) compared to non-human primates.

(6) An exploration of whether human tumor tissues exhibit mutations at this site and their potential influence on tumor progression, prognosis, or response to other treatment regimens would be a valuable addition to the study.

Response: We appreciate the reviewers' comments. Kindly see the discussion section, which has a perspective on the latter. In the SNP database, we found only one patient with Pro153-Ser153 mutation in the south Asian population. The data is included in Supplemental Fig 1.

(7) The study does not delve deeply into the mechanisms underlying human FasL's increase dsensitivity to plasmin, which is a critical aspect that warrants further investigation.

Response: We highly welcome reviewers' comments and suggestions. The revised manuscript, with significant new data, has now taken a deep dive into mechanistic FasL's increased sensitivity, including the use of FasL-engaging and plasmin cleavage-inhibiting antibodies. Other data sets include plasmin cleavage assays, the use of Fc-tag FasL human and murine forms, data with murine immune competent mice studies, the use of PD-L1 antibodies, and CAR-T bystander assays, to list a few. The revised manuscript is substantially strong and rationally provides mechanistic insight (with supporting data from multiple angles) into the human FasL increased sensitivity to plasmin.

(8) The presence of Ser153 in human FasL, an evolutionary distinction, raises questions about the safety of human FasL variants and plasmin inhibitors in tumor treatment. The study should address potential side effects and off-target effects on normal cells.

Response: We appreciate reviewers' comments. Kindly see the updated discussion section.

(9) While the manuscript suggests that understanding genetic variations in FasL could improve CAR-T cell-based immunotherapies, it does not explore how this insight could be practically applied. Further research in this direction would be beneficial.

Response: We appreciate the reviewers' comments and suggestions for CAR-T experiments. As per reviewer's suggestion we have tested the plasmin cleavage effect on FasL to block bystander signaling in heterogenous tumor conditions. Using anti-NaPi2b scFv (ovarian cancer-enriched sodium-dependent phosphate transporter receptor) we generated CAR-T cells and tested them a Fas bystander killing assay in heterogenous tumor settings by mixing NaPi2b⁺Fas⁺ and NaPi2b⁻Fas⁺tumor cells. The new set of data is provided in Figure 5h-l; kindly see the latest data and relevant text in the revised manuscript.

(10) In conclusion, the manuscript presents an interesting and potentially impactful study on the role of an evolutionary substitution in human FasL and its regulation by plasmin in cancer. With the suggested revisions, I believe this manuscript could make a significant contribution to the field of cancer immunotherapy.

Response: We highly welcome the strong, supportive, positive comment by the Reviewer that our manuscript presents an interesting and potentially impactful study on the role of an evolutionary substitution in human FasL and its regulation by plasmin in cancer. We also appreciate all the critical suggestions by the Reviewer, which has helped to improve the manuscript significantly.

Reviewer #3

The authors report the interesting finding that human FasL is cleaved by plasmin resulting in attenuation of its cytotoxic potential. They speculate that this could limit anti-tumoral T cell responses e.g. in CAR T cells therapies. The biochemical data concerning plasmin-mediated processing of human versus rhesus FasL, which is less efficiently processed, are convincing and straightforward. It should be clear stated that that FasL cleavage by plasmin has already been reported by Bajou et al. cited in the manuscript.

Response: We highly welcome the Reviewer's positive comment and sincerely appreciate the Reviewer's comment that "data related to human versus rhesus FasL, which is less efficiently processed, are convincing and straightforward." We also agree with the Reviewer that FasL cleavage by plasmin has already been reported by Bajou et al. PMID: 18835034. Indeed, the study published in cancer cells by Bajou et al. (PMID: 18835034), was very insightful for our whole differential huFasL regulation discovery in the context of evolutionary significance. The Bajou et al. study mainly focused on the endothelial context, and ours was focused on the cancer context. During the original submission, we were aware of the opposite effect in endothelial cells vs tumor cells. Hence, a significant amount of new data is provided in the revised manuscript to reconcile some of the observed differences in different cells/tissues.

The discussion of the functional data is however not fully convincing:

Response: We respectively with the Reviewer's comment that discussion was not fully convincing in original submission as only one short discussion paragraph was provided. In light of significant new data and as per reviewers' suggestion we have significantly expanded the discussion. We kindly request the Reviewer to read the revised manuscript.

Membrane FasL has much higher cytotoxic activity compared to proteolytically processed soluble FasL. Thus, an obvious interpretation would be that plasmin protects tumors from cytotoxic Fas activity by reducing the amount of the highly cytotoxic membrane FasL molecules. In this respect, I miss analysis of the effect of plasmin etc on the expression of human and rhesus membrane FasL. Ditto, I feel it is mandatory to evaluate the effect of plasmin on the cytotoxic action and processing of membrane FasL.

Response: We highly appreciate the Reviewer's comment. As per the Reviewer's suggestion, data on the cytotoxic action and processing of membrane FasL are included both using the cell lines (Fig 7g-h, Fig 7j-k, Supplemental Fig 10c-e), CAR-Ts cells which expressing membrane FasL (Fig 5h-l) and including T-cell-expressed FasL in immune-competent animal models (Fig 8 and 9).

The self-assembly of FasL is not only driven by the stalk region, dedicated by the authors as trimerization region. The TNF homology domain which is still functional present in plasmin-processed FasL is even more relevant for this as it is obvious from figure 2c.

Response: We fully agree with the Reviewer's comment that the self-assembly of FasL is not only driven by the stalk region. In the original submission, we did not run the SEC of the plasmin-cleaved form of human and rhesus FasL. Kindly see the revised data in Supplemental Fig 9a. Indeed, as the Reviewer predicted, plasmin cleaved FasL (146-281aa) formed trimers in the absence of stalk. Please also see the significant amount of new data in Fig 6 using human and murine soluble FasL, supporting that trimerization was independent of stalk but higher-order aggregation was higher in stalk-bearing huFasL.

It is known that oligomerization of soluble FasL trimers results in strong enhancement of their cytotoxic activity. Therefore, although soluble stalk-less and stalk-containing FasL variants may differ in their limited cytotoxic activity they still not reach the specific activity of oligomerized soluble FasL trimers which might better mimic the activity of memFasL.

Response: Again, we fully agree with the Reviewer's comment that the oligomerization of soluble FasL trimers would enhance cytotoxicity. Indeed, the latter is the case, and membrane huFasL and soluble large huFasL were equally effective if the Fas receptor oligomerization driven by them was similar. Kindly see the data in Supplemental Fig 10c and d (OVCAR-3) ovarian cancer part. As evident, both membrane FasL and soluble huFasL (huFasL^S) equally killed OVCAR-3 (Supplemental Fig 9c) cells and showed almost equal oligomerization of Fas and FADD (Supplemental Fig 9d. The smallest plasmin cleaved form of huFasL (huFasLST), which interferes with FasL self-aggregation but not trimerization (See Fig 6g, h and Supplemental Fig 10), showed no oligomerization of Fas receptor in OVCAR-3 lysates and was ineffective in killing them.

In view of the considerations above, the animal experiment shown in figure 2g and 2h possibly targets the secondary question of the different residual cytotoxic activity of different soluble FasL molecules instead the, in my opinion, more obvious question whether plasmin-dependent reduction of membrane FasL of T cells hinders the latter from tumor killing. Indeed, in accordance with the notion that all the soluble FasL are poorly active, when compared with relevant benchmarks, such as memFasL or oligomerized soluble FasL, the authors injected mice with 50 ug of their soluble FasL molecules every three days while other reported that already 5 ug of oligomerized soluble FasL kills mice within 3 hours (e.g. Jost et al., 2009). – High liver toxicity is considered as the hallmark of systemic CD95 activation!

Response: We appreciate the Reviewer's comment about the high general and liver toxicity of oligomerized soluble FasL in type-II cells, specifically if the XIAP expression is low, as described by Jost et al., 2009. However, we would like to point out that, in general, XIAP levels are highly elevated in most cultured cancer cells

(PMID: 16322751) that we used to generate tumors (PMID: 14749124) in mice as well as in human tumors. In our hands, we do see a high level of killing by 0.2-1.0 μ g FasL in vitro of cancer cells; however, as published earlier (PMID: 37838774), most tumors (which we inject every 3rd day) do not respond to huFasL injections below 40-50 μ g. We believe the discrepancy could also be due to endogenously expressed FasL near hepatocytes having high accessibility to tissue causing cytotoxicity while the exogenously added FasL is subjected to proteases and other steric regulation in the tumor microenvironment in addition to tumor-specific high XIAP expression.

It could also be discussed that soluble FasL may efficiently promote cell-death independent activities not requiring oligomerization by e.g. calcium signaling (Devel et al., 2022). I miss discussion of the existing literature concerning the activity of different forms of FasL including soluble FasL variants varying with respect to their "stalk content" and a stalk-related tendency to aggregate (without completeness e.g. Suda et al.; Berg et al.; Schneider et al.; Oyaizu et al.; Schulte et al.; Herrero et al.; O'Reilly et al).

Response: We appreciate the Reviewer's comment and agree that differential calcium signaling (Devel et al., 2022) could promote cell-death-independent activities. Upon the Reviewer's suggestion, the latter reference has been included in the discussion. In addition, we have provided context to other references listed by the Reviewer and have incorporated most of them appropriately according to the text in the manuscript.

Jost PJ, Grabow S, Gray D, McKenzie MD, Nachbur U, Huang DC, Bouillet P, Thomas HE, Borner C, Silke J, Strasser A, Kaufmann T. XIAP discriminates between type I and type II FAS-induced apoptosis. *Nature*. 2009 Aug 20;460(7258):1035-9. doi: 10.1038/nature08229.

Devel L, Guedeney N, Bregant S, Chowdhury A, Jean M, Legembre P. Role of metalloproteases in the CD95 signaling pathways. *Front Immunol*. 2022 Dec 5;13:1074099. doi: 10.3389/fimmu.2022.1074099.

Suda T, Tanaka M, Miwa K, Nagata S. Apoptosis of mouse naive T cells induced by recombinant soluble Fas ligand and activation-induced resistance to Fas ligand. *J Immunol*. 1996 Nov 1;157(9):3918-24. PMID: 8892623.

Berg D, Lehne M, Müller N, Siegmund D, Münkler S, Sebald W, Pfizenmaier K, Wajant H. Enforced covalent trimerization increases the activity of the TNF ligand family members TRAIL and CD95L. *Cell Death Differ*. 2007 Dec;14(12):2021-34.

Schneider P, Holler N, Bodmer JL, Hahne M, Frei K, Fontana A, Tschopp J. Conversion of membrane-bound Fas (CD95) ligand to its soluble form is associated with downregulation of its proapoptotic activity and loss of liver toxicity. *J Exp Med*. 1998 Apr 20;187(8):1205-13.

Oyaizu N, Kayagaki N, Yagita H, Pahwa S, Ikawa Y. Requirement of cell-cell contact in the induction of Jurkat T cell apoptosis: the membrane-anchored but not soluble form of FasL can trigger anti-CD3-induced apoptosis in Jurkat T cells. *Biochem Biophys Res Commun*. 1997 Sep 18;238(2):670-5.

Schulte M, Reiss K, Lettau M, Maretzky T, Ludwig A, Hartmann D, de Strooper B, Janssen O, Saftig P. ADAM10 regulates FasL cell surface expression and modulates FasL-induced cytotoxicity and activation-induced cell death. *Cell Death Differ*. 2007 May;14(5):1040-9.

Herrero R, Kajikawa O, Matute-Bello G, Wang Y, Hagimoto N, Mongovin S, Wong V, Park DR, Brot N, Heinecke JW, Rosen H, Goodman RB, Fu X, Martin TR. The biological activity of FasL in human and mouse lungs is determined by the structure of its stalk region. *J Clin Invest*. 2011 Mar;121(3):1174-90.

O'Reilly LA, Tai L, Lee L, Kruse EA, Grabow S, Fairlie WD, Haynes NM, Tarlinton DM, Zhang JG, Belz GT, Smyth MJ, Bouillet P, Robb L, Strasser A. Membrane-bound Fas ligand only is essential for Fas-induced apoptosis. *Nature*. 2009 Oct 1;461(7264):659-63.

Response: We appreciate the Reviewer's suggestion to comment on the given references. Upon the Reviewer's suggestion, most of the references listed by the Reviewer and have been incorporated appropriately according to the text in the manuscript.

REVIEWER COMMENTS

Reviewer #1 (Remarks to the Author):

The authors have provided a detailed response to my earlier comments. The manuscript is significantly longer as a result. I am for the most part satisfied with the authors responses to my questions and that FasL contain a unique plasmin cleavage site that is present in human but not non-human FasL. A couple of questions remain:

Fig 3d: the authors claim that the 144AA145 mutation completely abolished the 31kD fragment release. However, there is a band migrating slightly faster than 31kD (~25kD) that the authors state to be "non-specific". How can the authors be sure of this and that this is not a band produced from another cleavage site?

Response: We appreciate reviewer's comment, and concern. As we are making so many mutants, we must express multiple batches of IgG1Fc-conjugated FasL. Hence, we do notice that in some expressed protein batches a non-specific band even in untreated samples under 31KDa but not in others. As an example, please see Fig 3c. untreated vs Fig 3d untreated (the revised submitted manuscript submitted earlier).

To further address reviewer concern, we again used IgG1Fc-conjugated FasL WT and RK (R144A, R145A) mutants from 3 different expression sets. The 3 different sets of proteins were purified by 3 different people in our lab, and we re-did the assay again in 3 sets with varied plasmin conc. and incubation times including O/N incubation of low plasmin conc. (rather than just 1 set which was given in revised submission). Kindly see the attached data here for you. We have also provided saturated blot (set-1 bottom) and overlay of PVDF membrane with ladder (size marker) in set-2 bottom and set-3 bottom.

Kindly see, 31KDa band in all 3 sets (compare left part of gel to the right part). No such band is evident in any of the sets. Also, the faint band under 31KDa band is not evident in RK mutant cleavage in set-1 or set-2 even after very long saturation (set-1) and overnight incubation (set-2). In set 3, the non-specific faint band (below 31KDa) is evident even in untreated hence it is in all samples with varied intensity. Thus, the data is highly conclusive (n=4 total) that RK mutant is resistant to plasmin cleavage and does not release plasmin cleavage band at FasL site. As per reviewers' suggestion we have replaced data in 3d with set-1 and set 2 is added as supplemental Fig 6c.

It iWestern blots (in general). The description of the western blots is insufficient. At various times, the authors conducted western blots using "non-reducing and partly denatured gels". What does this mean? Samples are either denatured or not.

Response: We appreciate reviewer's comment and in agreement with reviewer have updated the text with non-reducing (with 60% reduced SDS conc) gels.

Reviewer #2 (Remarks to the Author):

I'm glad to receive the authors' point-by-point response. After looking at the new data they provided, I've found several areas that could be improved.

1. Issues with CD3 Staining and Gating in Fig8 i/j: There seem to be some problems with the CD3 staining and gating in Fig8 i/j. In the FACS plots, the cells shown should include both CD3-positive and -negative populations. In the secondary group (I assume it's the control group stained with CD3 and FasL antibodies), all cells are CD3-negative. In the IgG1 Ctrl group, there are both CD3-positive and -negative populations. However, in the Avelumab IgG4 group and Avelumab IgG4 + Nok2h scFv group, almost all cells are CD3-positive. I'm curious about the reasons for these differences among the last three groups. Why are the last two groups mainly composed of CD3-positive T cells, while the IgG1 Ctrl group has two distinct populations? And in the IgG1 Ctrl group, why isn't the separation between CD3-positive and -negative cells clear? If all groups are gated on the CD3+ T-cell population, how does the CD3-negative population in the IgG1 Ctrl group exist?

Response: We appreciate reviewers' very important comment and appreciate him/her for critically looking into figures (Thank you!). In the revised manuscript, we mistakenly provided the IgG1 ctrl from another manuscript that we are working on side by side. If reviewer would look carefully, the right IgG1 Ctrl with separated CD3-positive population was provided in one set of Supplemental Fig 13 for both MC38 and ID8 sets. That set was supposed to be in main Figure 8i/j. We apologize for confusion.

The correct IgG1 group is added, kindly see updated manuscript now.

We would like to point out that although the trend was clear in provided revised manuscript avelumab and avelumab+ Nok2h group to support the hypothesis, we were also not very pleased about the population not being very clearly enriched. We believe that the not so very clear population was because of miltenyi magnetic beads being older (stored for a while). Hence, we were repeating the whole experiments with brand new magnetic beads even to acquire the missing n=3 data for Fig.9b (which reviewer have also requested in point 5). Hence, a brand-new data set with clear cell populations is provided. Kindly see both updated Fig 8i/j and Fig 9b.

Yes, we do fully agree with the reviewer that in the Avelumab IgG4 group and Avelumab IgG4 + Nok2h scFv group were very supportive of our hypothesis (having almost all cells are CD3-positive and enriched) while there were two populations in the IgG1 Ctrl group.

Based on our previous experiences (DOI: published but cannot share as it is blind Review), the immune population from syngeneic tumors in general are very heterogenous, hence upon previous reviewers suggestion that time, we enriched the T-cells (CD3 or CD4 or CD8+) prior to FACS using enrichment magnetic beads to get clear FACs data as published earlier by us and many others. The methods about the latter were also indicated in the material methods section. We have updated the latter in text related to figure, figure legends and figure schematics (Fig. 8h, Fig. 9a) to avoid any confusion. The updated version is highlighted in blue color text (and blue line on left).

Also, the secondary group represent secondary antibody alone as negative control for the background signal. We have updated properly in the figure to avoid confusion.

2.Problems with CD3 Staining and Gating in Fig9b: Similar issues exist in the CD3 staining and gating of Fig9b. There are clearly two populations of CD3-stained cells. But with the current +- shaped gate, the CD3-positive population is further divided into two parts (the left part is classified into the negative group), making the CD3-negative cells appear as two distinct populations. Although the authors set the positive gate based on the Secondary group, this gating method doesn't work well in this case. Similarly, the IFN gate also splits an obvious cell population into two, which is not ideal. I suggest the authors use an irregular gate to solve this problem.

Response: We appreciate reviewers' comments and agree the gate is slight right, which gives the impression that CD8-positive population or IFN population is divided into two. As per reviewer suggestion, if we update the gate, the second population would disappear, and data will be clear with clear population. However, as said in point 1 (and to point 5) we were repeating this whole experiment and Fig 8 FACs experiment to get n=3. Hence new FACS figure with clear CD8 population is provided. Also, the In Fig9b the population is CD8+, (not CD3, reviewer typo) enriched and was used in FACs to analyze double positive (CD8+, IFN +) population. The updated version is highlighted in blue color text (and blue line on left).

3.Scarcity of CD3-Negative Populations in Fig8 i/j and Fig9b: In Fig8 i/j and Fig9b, the CD3-negative populations are surprisingly small. I'm wondering where the other immune cells have gone. It would be great if the authors could explain this.

Response: Kindly see response to point 1 and point 2. As stated, and published by us and various groups, we have enriched for T-cells (using magnetic beads) to avoid other immune cells prior FACs, hence the population are small. Kindly the latter was also stated in material and methods and has also been now updated in Figure Schematics (Fig. 8h, Fig. 9a) and Figure legends. The updated version is highlighted in blue color text (and blue line on left).

4.Difference in FasL-Negative Populations in Fig8 i/j: In Fig8 i/j, the FasL-negative population in the IgG1 Ctrl group looks quite different from the other two groups. Could the authors provide an explanation for this difference? It might help us better understand the data.

Response: Once again, we apologize for IgG1 Ctrl group confusion, kindly see the response to point 1.

5.Suggestion for Fig9b: For the FACS analysis in Fig9b, I suggest the authors create a statistical graph similar to Fig8k. This would make the data more intuitive and easier to compare.

Response: We appreciate reviewers' forward-thinking comments. We were repeating this experiment in case it would be requested. Kindly see n=3 and statistical graph is added in Fig 9c. The updated version is highlighted in blue color text (and blue line on left).

6.Typo in FACS Staining Controls: In Fig8i/j and Fig9b, the FACS staining control is labeled as "secondary" in some places, but it should be "secondary". Please correct this typo to avoid confusion.

Response: We appreciate reviewers' comments, and we have updated it as secondary (2° Ab) only in all places.

7.Inconsistency in Antibody Description: The authors' team treated the mice with the anti-PD-L1 (avelumab-IgG4) antibody. However, in the main text, it's written as "PD-L1 treated". Usually, this kind of expression implies treatment with the PD-L1 protein, not the antibody. It would be better to use a more accurate description to avoid misunderstandings

Response: We appreciate reviewers' critical suggestion. We have updated "PD-L1 treated" with "anti-PD-L1 treated" throughout the manuscript. The updated version is highlighted in blue color text (and blue line on left).